# Large-scale shift in the structure of a kelp forest ecosystem co-occurs with an epizootic and marine heatwave

Meredith L. McPherson [1✉], Dennis J. I. Finger [2], Henry F. Houskeeper [1,3], Tom W. Bell [4,5], Mark H. Carr[6], Laura Rogers-Bennett [7] & Raphael M. Kudela [1]

Climate change is responsible for increased frequency, intensity, and duration of extreme events, such as marine heatwaves (MHWs). Within eastern boundary current systems, MHWs have profound impacts on temperature-nutrient dynamics that drive primary productivity. Bull kelp (*Nereocystis luetkeana*) forests, a vital nearshore habitat, experienced unprecedented losses along 350 km of coastline in northern California beginning in 2014 and continuing through 2019. These losses have had devastating consequences to northern California communities, economies, and fisheries. Using a suite of in situ and satellite-derived data, we demonstrate that the abrupt ecosystem shift initiated by a multi-year MHW was preceded by declines in keystone predator population densities. We show strong evidence that northern California kelp forests, while temporally dynamic, were historically resilient to fluctuating environmental conditions, even in the absence of key top predators, but that a series of coupled environmental and biological shifts between 2014 and 2016 resulted in the formation of a persistent, altered ecosystem state with low primary productivity. Based on our findings, we recommend the implementation of ecosystem-based and adaptive management strategies, such as (1) monitoring the status of key ecosystem attributes: kelp distribution and abundance, and densities of sea urchins and their predators, (2) developing management responses to threshold levels of these attributes, and (3) creating quantitative restoration suitability indices for informing kelp restoration efforts.

[1] Department of Ocean Sciences, University of California Santa Cruz, Santa Cruz, CA, USA. [2] Department of Environmental Science, Policy, and Management, University of California Berkeley, Berkeley, CA, USA. [3] Department of Geography, University of California Los Angeles, Los Angeles, CA, USA. [4] Department of Applied Ocean Physics and Engineering, Woods Hole Oceanographic Institution, Woods Hole, MA, USA. [5] Earth Research Institute, University of California Santa Barbara, Santa Barbara, CA, USA. [6] Department of Ecology and Evolutionary Biology, University of California Santa Cruz, Santa Cruz, CA, USA. [7] Coastal Marine Science Institute, Karen C. Drayer Wildlife Health Center, University of California Davis and California Department of Fish and Wildlife, Bodega Marine Laboratory, Bodega Bay, CA, USA. ✉email: mmcpher1@ucsc.edu

Coastal marine ecosystem response to climate change in the 21st century is predicted to manifest in various ways, including through habitat contraction, species range shifts, and losses of biodiversity and functionality[1]. These responses can manifest through both long-term gradual changes and more episodic events[2,3]. However, it can be difficult to distinguish the impacts of gradual (e.g., increasing mean temperatures) and irregular (e.g., increasing storm frequency) climate-induced shifts from changes in underlying naturally stochastic events (e.g., El Niño Southern Oscillation (ENSO)). One such example is the ocean warming phenomenon of marine heatwaves (MHWs). Global oceanic and atmospheric drivers influence the regional frequency, duration, and intensity of MHWs[4,5], all of which are increasing[4,6]. In eastern boundary current ecosystems, such as the California Current, MHWs are highly correlated to changes in nutrient availability given the strong correlation between temperature and nutrients[7,8] (e.g., anomalously high sea surface temperature (SST), low nitrate concentration $[NO_3]$). MHWs can have notable impacts on coastal ecosystems, such as seagrass beds[9], coral reefs[10] and kelp forests[11], and especially on the foundation species and ecosystem engineers (e.g., seagrasses, corals, kelps) that define these systems. Furthermore, the specific impacts of climate-induced changes to these habitat-forming sessile organisms via the coupled impacts of regional non-climate change human influences and species thermal tolerance, greatly increase their vulnerability relative to mobile species[12].

Canopy-forming kelp species (Order: *Laminariales*) thrive along temperate rocky coastlines, and are the foundation of productive and species-rich ecosystems that generate a diversity of provisioning, regulating, and supporting ecosystem services[13]. Despite the high regional variation, global rates of kelp forest loss have generally increased over the last 20 years due to a combination of short and long-term anthropogenic influences[14]. Furthermore, the influences of ocean warming on kelp forest systems have been observed across nearly every ocean basin[11]. Intense warming has occurred in localized coastal regions of Western Australia[15,16], the Tasman Sea Region[17], New Zealand[18], Baja California[19,20], Nova Scotia, Canada[21], and northern California[22]. While the direct ecological implications of MHWs on kelp forests are not fully understood, MHWs can alter ecosystem structure and functioning via shifts in kelp community species composition[15,23,24] leading to dramatic ecosystem shifts from healthy forest to algal turf reefs[25] or sea urchin barrens[26]. These shifts between alternative stable states of these kelp forests often reflect cascading interactions across trophic levels through bottom-up (i.e., environmental influences on kelps) and top-down[27–29] (i.e., changes in predator control of grazers) processes.

Along the coast of northern California (Fig. 1a; Mendocino and Sonoma Counties), anomalously warm seawater temperatures persisted from 2014 to 2016 (Fig. 2b) caused by an ocean warming event (termed 'the blob') associated with global climate change and a strong El Niño event[30] (collectively referred to as the NE Pacific MHW). The year prior (2013), the onset of a sea star wasting syndrome (SSWS) epidemic caused dramatic population declines in multiple species of sea stars including the sunflower star, *Pycnopodia helianthoides*, along the entire northeast Pacific coastline (Fig. 2d)[22,31]. The sunflower star was the primary predator of sea urchins in northern California since the historic extirpation of sea otters (*Enhydra lutris*)[32]. Aligned with these events, forests of bull kelp, *Nereocystis luetkeana*, exhibited an unprecedented ecosystem shift from healthy forests to 'urchin barrens' devoid of macroalgae along more than 350 km of coastline[22]. Prior to this regime shift, the forested ecosystem likely persisted because the cool, nutrient-rich waters that fueled kelp production and food availability to urchins[33,34] was balanced by top-down urchin predation by the sunflower star[35].

In contrast to giant kelp, which can live for many years and continuously produce new reproductive and vegetative fronds, bull kelp's annual life history is limited to the production of a single stipe and its reproductive blades in its lifetime[36,37]. As such, mechanisms for spore dispersal are limited to a narrow window between the maturity of the kelp and the onset of fall and winter storms, which usually dislodge adult bull kelp from the substrate (except in areas protected from wave energy). These factors lead to high spatial and temporal variability in the distribution and abundance of the surface canopy that can be observed through remote sensing techniques.

Satellite imagery provides a unique perspective on how surface canopy-forming kelps respond to both acute climate manifestations and in situ biological trends leading to ecosystem phase shifts and can compensate for the scarcity of historical kelp data in northern California. This dataset precedes the recent influence of the NE Pacific MHW allowing us to explore the contribution of environmental and biological factors on short and long-term trends of kelp canopy coverage. Using a 34-year time series of kelp canopy coverage derived from United States Geological Survey (USGS) Landsat imagery in combination with large-scale, local-scale, and biological drivers, we infer that historically declining predator densities might have laid the groundwork for the observed ecosystem phase shift in northern California initiated by a multi-year MHW. The persistent lack of kelp from 2014 to 2019 does not appear to be the result of unfavorable environmental conditions alone, but by a combination of unfavorable conditions for kelp productivity (related to warm SST and low nutrients) and conditions favorable for the persistence of urchin populations, including recruitment and low rates of mortality stemming from the absence of predators, disease, and starvation. Thus, while fluctuating environmental conditions occurred throughout the past three decades, the combination of abrupt changes in environmental and biological conditions likely hindered the ability of the ecosystem to recover as it had over the past three decades. This work provides context for monitoring biological trends and environmental response in surface canopy-forming kelp forest ecosystems globally where satellite monitoring can be applied. These techniques are becoming increasingly important for designing adaptive management strategies to mitigate the impacts of long-term and abrupt environmental stressors.

## Results

Northern California bull kelp displayed a dynamic inter-annual pattern of canopy coverage[38,39] across the 34-year satellite-derived record prior to 2014 (Fig. 1b). The onset of NE Pacific MHW and prior mass mortality of sunflower stars via SSWS coincided with dramatically reduced kelp canopy area in 2014 (Figs. 2d and 4a). Mean SST anomalies during the MHW event from 2014 to 2015 were ~2 standard deviations warmer, with extreme SST anomalies reaching 3 to 4 standard deviations above the long-term mean distribution (Fig. 2b). Anomalously cool, nutrient-replete conditions ideal for bull kelp growth were observed in 2012 and 2013 (Fig. 2c), but canopy area fell dramatically in 2014 and has remained suppressed through 2019 even though environmental conditions became more favorable to kelp (Figs. 1b and 2d). The spatial range of bull kelp was also compressed across the last three decades (Fig. 1b). Specifically, the meridional range of kelp narrowed with complete disappearance within the northern-most region of the study area (north of Fort Bragg) after 2008 and in the southern-most region after 2012. The northern and southern portions of the historical kelp canopy distribution within our study area are regions characterized by sandy sediment[40] (poor substrate for kelp spore settlement[36]),

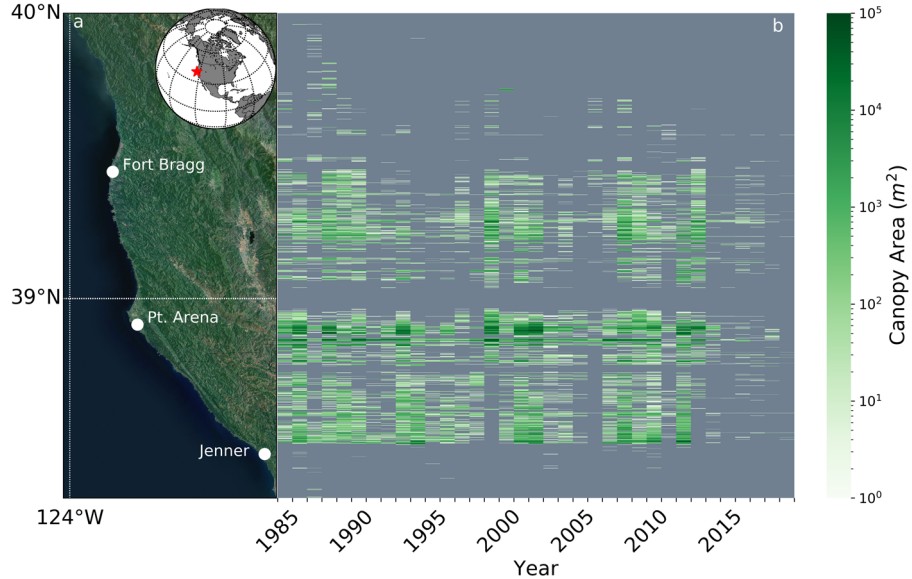

**Fig. 1 Spatial and temporal variability of bull kelp canopy area in northern California from 1985 to 2019. a** Sonoma and Mendocino county region of study and SST domain (Esri World Imagery – Esri, CGIAR, USGS HERE, Garmin, FAO, METI/NASA, EPA, Earthstar Geographics) and inset with a global map indicating the northern California region with a red star, **b** annual timeseries heatmap of kelp canopy summed within 90 m latitudinal bins. Esri. "World Imagery" [basemap]. Scale Not Given. "World Imagery Map". December 12, 2009. https://www.arcgis.com/home/item.html?id=10df2279f9684e4a9f6a7f08febac2a9. (Jan 26, 2021).

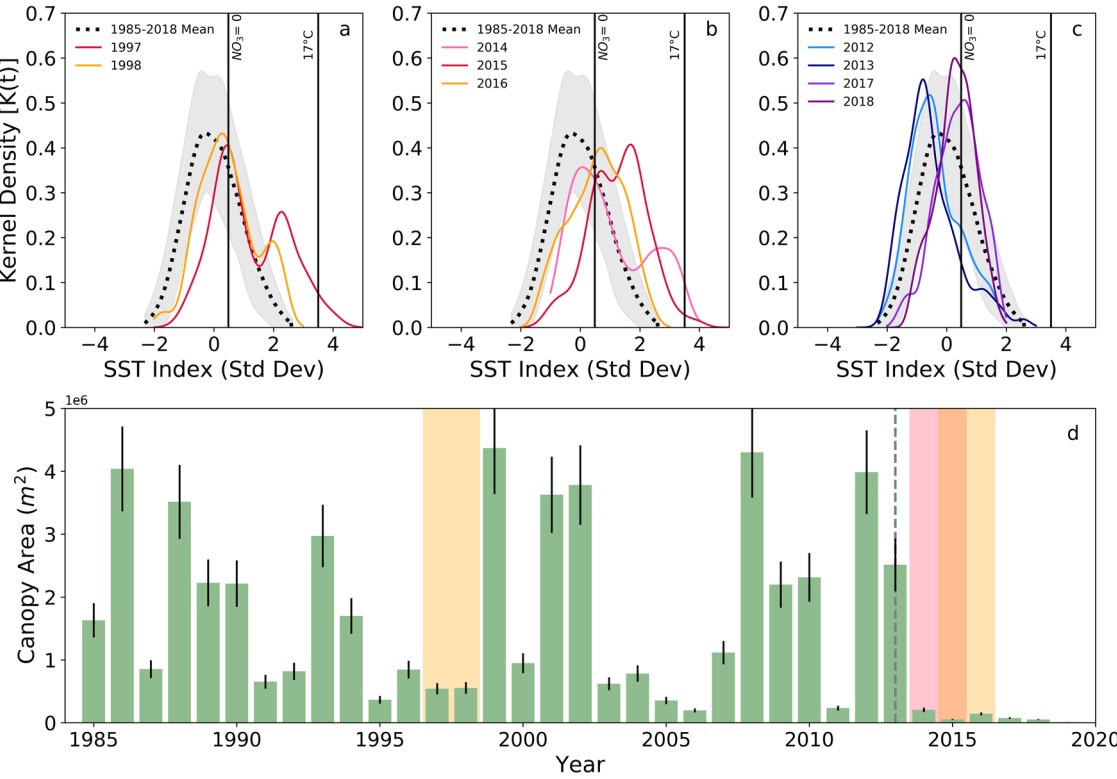

**Fig. 2 SST distribution and kelp canopy area in northern California during prominent El Niño and MHW events from 1985 to 2019.** Kernel density functions for SST anomalies during **a** the 1997/1998 El Niño, **b** the 2014–2015 NE Pacific MHW event (i.e., 'blob' and El Niño), and **c** relatively normal conditions before and after the MHW event (2012/2013 and 2018/2019, respectively). Shaded gray areas (**a–c**) represent ±1 SD from the long-term mean SST index. Solid black lines (**a–c**) represent the physiological threshold for bull kelp at 17 °C[65] ($\sigma_{1985-2019} = 3.5$) and the $NO_3$ deplete ($NO_3 = 0$) threshold ($\sigma_{1985-2019} = 0.48$). **d** Kelp canopy coverage through time with relevant oceanographic and biological events overlaid onto the timeseries as follows: a shaded yellow bar during the 1997/1998 El Niño; a shaded red bar during the 2014/2015 'blob'; a shaded orange bar during the overlapping 'blob' and 2015/2016 El Niño; a shaded yellow bar during the 2015/2016 El Niño; and a dashed gray line in which SSWD in sunflower stars was first observed in 2013. Annual error estimates (black error bars) for kelp canopy area were determined using the normalized root mean square error (NRMSE) between CDFW arial flyover surveys and USGS Landsat imagery[66].

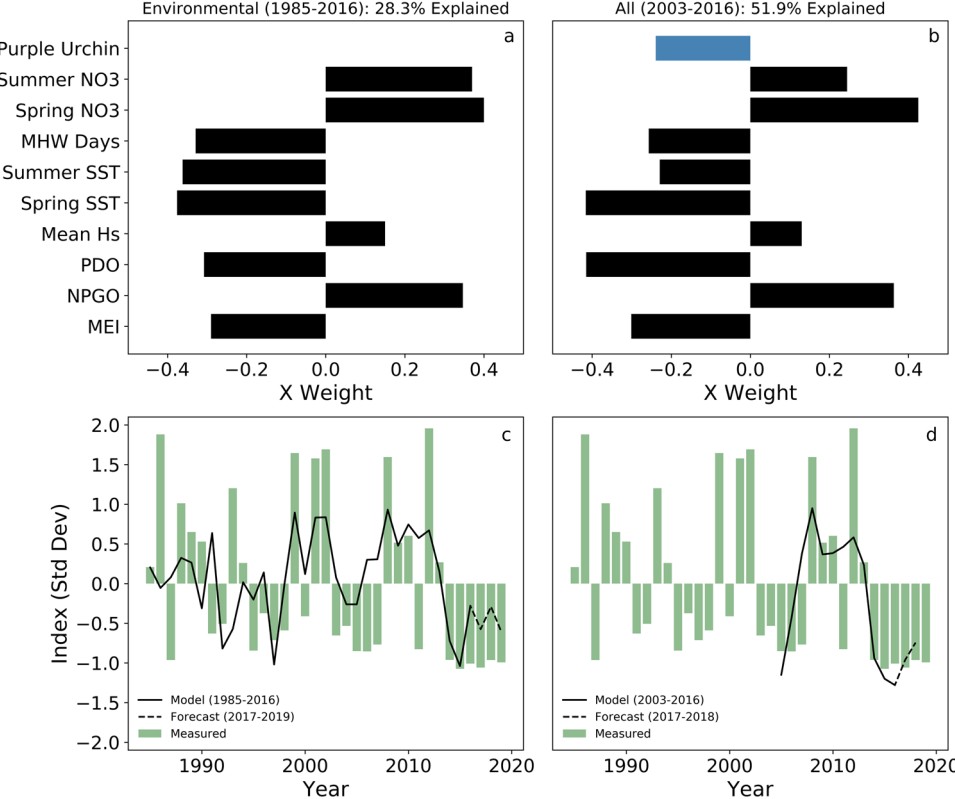

**Fig. 3 Results of Partial Least Squares Regression analysis for environmental and biological drivers of kelp canopy area from 1985 to 2019.**
Component 1 partial least squares regression (PLSR) x weights (top row) from environmental indices across 1985 to 2016 (**a**) and both environmental and biological indices from 2003 to 2016 (**b**). PLSR models and forecasts using all components overlaid on satellite-derived kelp canopy (**c, d**). See supplementary data for detailed PLSR results (Supplementary Table 1). Predictor variable acronyms are as follows: purple urchin density—'Purple Urchin'; seasonal nitrate concentrations—'Summer NO$_3$' and 'Spring NO$_3$'; marine heatwave days—'MHW Days'; seasonal sea surface temperature— 'Summer SST' and 'Spring SST'; mean significant wave height—'Mean H$_s$'; Pacific Decadal Oscillation—'PDO'; North Pacific Gyre Oscillation—'NPGO'; Multivariate El Niño/Southern Oscillation Index—'MEI'. See Methods for a detailed description of how each environmental variable influence kelp canopy dynamics.

resulting in sparser and patchier distribution than the rockier coastline between Fort Bragg and Jenner prior to the NE Pacific MHW. In addition to range reductions, the total loss of canopy area was spatially apparent beginning in 2014 when sparse, patchy conditions began dominating historically dense regions of the coastline (e.g., Point Arena; Fig. 1b).

Spatial and temporal variability were clearly apparent in the satellite-derived northern California kelp record. Based on results from partial least squares regression (PLSR) analyses, patterns of change in these systems were generally described by co-varying mechanistic drivers of environmental and biological processes (Fig. 3a and b; Supplementary Table 1) including nitrate (NO$_3$) availability, SST (relevant to [NO$_3$] and physiological temperature thresholds), large-scale ocean-atmospheric forcing (e.g., Multivariate El Niño Southern Oscillation Index (MEI), North Pacific Gyre Oscillation (NPGO), Pacific Decadal Oscillation (PDO); all which drive local patterns of [NO$_3$] and SST), the seasonal timing of swell (significant wave height (H$_s$), which influences spore dispersal), and grazer (purple sea urchin) abundances.

Our results revealed that including grazer dynamics in a predictive model more accurately represented sustained low kelp biomass after environmental perturbations from the NE Pacific MHW than the same predictive model with grazer abundance omitted. Environmental drivers correctly represented bull kelp response to low NO$_3$ and high SST conditions across the NE Pacific MHW regardless of whether the event was included in the temporal representation of the PLSR model (Fig. 3c solid black line) and forecast (Fig. 3c dashed black line) results. Forecasted

model results using only environmental drivers indicated that bull kelp was expected to partially recover in 2017 once SST and NO$_3$ concentrations rebounded from extreme anomalous conditions (Fig. 3c). However, full bull kelp recovery in the environmental-only model forecasts may be hindered by NO$_3$ concentrations that remained below the long-term average after the NE Pacific MHW. Including urchin (grazer) dynamics in the PLSR analysis show that low kelp canopy biomass conditions persist regardless of the anticipated effects of environmental drivers to kelp recovery (Fig. 3d). Furthermore, simulating a recovery of environmental drivers to the long-term climatological mean across 2017-2019 suggested that high urchin conditions disproportionately suppress kelp relative to environmental drivers (Supplementary Fig. 1).

Northern California kelp has historically responded to fluctuations in temperature extremes such as the 1997/1998 El Niño event (depicted in Fig. 4a by the sharp peak in the 1997 annual number of MHW days) but has been resilient to widespread collapse. The substantial declines (well below the long-term mean) in sunflower stars evident in 2013 (Fig. 4c; Supplementary Fig. 2 and Supplementary Table 2; slope = −0.23 m$^{-2}$ yr$^{-1}$; $p$ = 0.01) set the stage for a system-wide phase shift into an urchin barren state initiated by the NE Pacific MHW event[22]. Increases in purple urchin densities lagged anomalously low sunflower star densities by one year, spurred by reduced top-down forcing by sunflower stars[41] and a large purple urchin larval recruitment event in 2014[42] (Fig. 4). Bull kelp and sunflower stars also exhibited stepwise functions across the MHW event (Figs. 4b

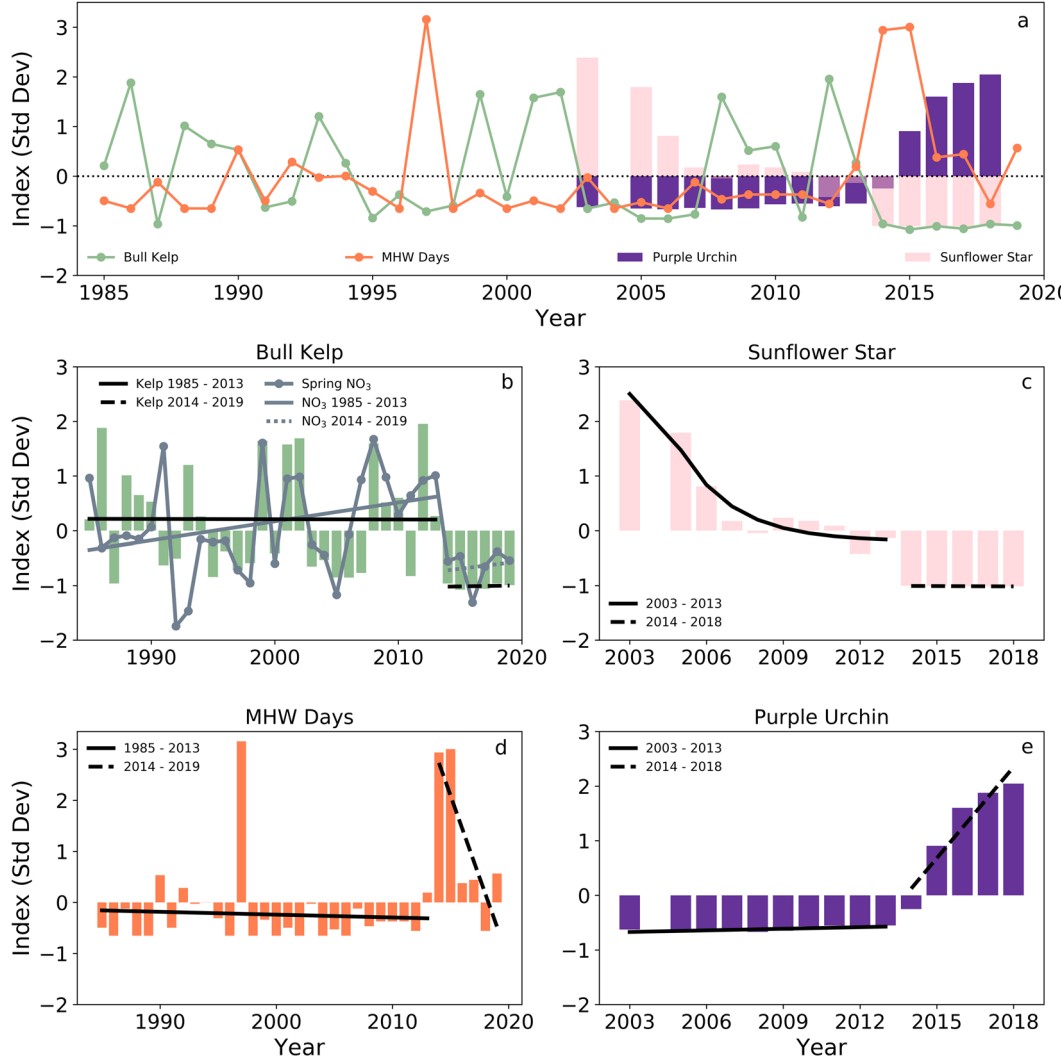

**Fig. 4 Temporal trends of important environmental and biological drivers of ecosystem change in northern California kelp forests.** Standardized indices of (**a**) bull kelp canopy coverage, MHW days, purple urchin density, and sunflower star density where data are available. Standardized indices overlaid with Ordinary Least Square Regression (OLSR) fits (except in 4c 2003–2013 where a second degree polynomial LSR is applied) prior to and after the NE Pacific MHW for (**b**) bull kelp canopy coverage and nitrate concentration, (**c**) sunflower star density, (**d**) MHW days, and (**e**) purple urchin density. See supplementary data for detailed LSR results and error statistics (Supplementary Fig. 2 and Supplementary Table 2).

and c). Absolute mean densities for both organisms stabilized close to zero, represented by the anomalously low index values between 2013 and 2018 for sunflower stars and between 2014 and 2019 for bull kelp (Fig. 4c; Supplementary Fig. 2 and Supplementary Table 2). Despite temperature anomalies returning to near normal distributions (Fig. 2c) and spring nitrate concentrations rebounding slightly from minimums observed in 2016 (Fig. 4b), barren conditions likely persist because of a widespread shift in purple urchin foraging behavior[34] (Fig. 4e; slope $= 3.1 \pm 0.67$ m$^{-2}$ yr$^{-1}$; $p = 0.02$) and sustained high densities (mean $= 14.8 \pm 8.3$ m$^{-2}$) hinder a reversal back to a healthy kelp forest state (Fig. 4e).

## Discussion

Northern California kelp forests experienced environmental and biological perturbations that likely resulted from the combined effects of (1) the absence of top-down control on urchin populations during and after the NE Pacific MHW (Fig. 4c), (2) abrupt and persistent shifts in SST and nutrient conditions across the NE Pacific MHW that were beyond the physiological thresholds of

optimum bull kelp growth and reproduction, and (3) an eruption in the population and grazing intensity of the herbivorous purple sea urchin. Previous work on the dynamics of marine and terrestrial ecosystem shifts sheds light on how these transitions in northern California were initiated by environmental events[35,43,44] and preceded by low ecosystem resilience.

Co-varying environmental parameters, including SST and nitrate concentrations, historically maintained fluctuating yet stable long-term trends of bull kelp conditions in northern California (Fig. 4d; $p > 0.05$). However, differences in the expression of kelp forest canopy dynamics between two foundational kelp genera across the NE Pacific MHW highlights that the annual life cycle of bull kelp makes them particularly sensitive to acute stressors[36], such as MHWs and prolonged nutrient deplete conditions (Fig. 2 a–c). This is evidenced by the fact that the stepwise decline in northern California bull kelp canopy area across the NE Pacific MHW was not observed in giant kelp (*Macrocystis pyrifera*) canopy biomass at a regional scale in southern California[19] and northern Baja California[19,20,24]. These observations suggest that giant kelp responded strongly to the NE Pacific MHW as a function of the genera's physiological

temperature threshold and latitudinal gradients in SST magnitudes[19], most likely because they were near their southern range and thermal limit in the northern hemisphere (Baja California, Mexico to Aleutian Islands, AK). In contrast, bull kelp forests in our study area, which lie in the middle of their distribution (Point Conception, CA to Unimak Island, AK), did not experience patchy spatial and temporal recovery after the onset of the NE Pacific MHW but maintained very low biomass conditions between 2014 and 2019, perhaps exacerbated by low propagule pressure resulting from patchy, sparse kelp densities and an annual life history strategy[36]. Furthermore, sea urchin dynamics differed between northern California and southern/Baja California. Increases in crowned sea urchin density (and decreases in invertebrate species richness) in localized areas of the Baja region indicate enhanced grazing pressure, in addition to temperature stress, may have occurred but not on a regional scale[20].

Regional-scale sea urchin larval recruitment dynamics are associated with large-scale environmental drivers and subsequent population dynamics[42]. Anomalously high larval recruitment was observed in Fort Bragg, CA, peaking in 2015 and increased larval settlement appeared to be correlated with juvenile and adult urchin densities during the NE Pacific MHW. Although there are no reliable in situ data available for sea urchin densities prior to 2003 from the northern California region, there are purple urchin settlement data as early as 1990 from Fort Bragg (1990–2016[42]) and Westport, Pt. Cabrillo, and Pt. Arena (1990–1993[45]). These settlement records show there was anomalously high larval settlement from 1993 to 1994 and 1998. Despite bull kelp canopy area being anomalously low between 1995 and 1998 (Fig. 4b), there is no evidence of complete kelp forest collapse or ecosystem shift during that period and no way to verify that high juvenile urchin densities coincided with the high larval densities. Furthermore, anomalously low kelp conditions do not always follow or co-occur with sea urchin larval settlement events (e.g., Fig. 4b 2003– 2007; Okamoto et al.[42], Fig. 3a). Given the positive relationship between SST and larval settlement in northern California[42] and predictions of more frequent and/or severe MHW[46], restoration efforts in northern California would benefit from a greater understanding of localized sea urchin population dynamics in that region.

Observed historical declines in predator diversity is reflective of a reduction in ecosystem resilience. The sequence of biological events in northern California, beginning with the extirpation of sea otters in the 1800s, appears to have reduced the resilience of kelp ecosystems across the entire region[43,47]. Throughout California, a suite of predators (e.g., sunflower stars, sea otters, California Sheephead fish[34]), and their complementary effects, play an essential role in maintaining stable forested states by enhancing resiliency via size-dependent predation[35], even when environmental perturbations occur. In sea urchin barrens, urchins are starved and lack energetic value to predators with high metabolic rates[48]. Moreover, behaviorally mediated predators often track changes in the distribution of profitable prey, which further complicates implications for recovery. Urchin barrens are characterized by a low urchin gonad index; because the gonads are what urchin predators target and consume, urchin barren states potentially limit increased predation of this particular prey[49]. The high urchin densities observed in northern California have induced starvation conditions and reduced nutritional value[49]. In addition, since the overall ecosystem biodiversity of urchin barrens is severely reduced[50], opportunities for predator (otters, sunflower stars, etc.) recovery is diminished. Ecosystem recovery is further limited by evidence that the effects on prey can lag behind the recovery of a predator[51].

Despite potential limitations of urchin barrens on predator recovery, the reintroduction of sea otter populations into urchin barrens has resulted in phase shifts back to forested states in some locations (e.g., Aleutian Islands[26]). It is unclear from this analysis what future phase state dynamics will occur with the reintroduction of a top predator given the strong potential that this urchin barren constitutes a kelp forest alternative stable state[26]. Although we refer to the recent wide-spread kelp forest loss as a phase shift and cannot currently provide proof of a true kelp forest alternative stable state, Filbee-Dexter and Scheibling[26] argues that in most cases the formation of urchin barrens can be regarded as such. Considering that the dynamics of the widespread urchin barren in northern California has similar patterns to other urchin barrens, hysteresis (discontinuous phase shift) and strong positive feedbacks may maintain the current state for a prolonged period of time.

Whatever mechanisms of system-wide resilience that existed prior to the complete loss of sunflower stars in 2014 were eliminated by its removal[22,24,29]. Though recovery of other sea star species has been observed across the NE Pacific coastline, the sunflower star remains locally extinct from kelp forest and intertidal ecosystems along the entire region. Evidence suggests that the pathogen associated with SSWS in the sunflower star was not temperature dependent, nor responsible for disease observed in other asteroids throughout the region[52]. This may explain why recovery for sunflower stars across the region, and in turn kelp forest recovery in northern California, remain absent despite temperature and nutrient conditions recovering slightly in 2017. Furthermore, the clear phase shift observed in other biological and environmental conditions in northern California (Fig. 4b, d, and e), such as sunflower star populations, began to decline well before the NE Pacific MHW in a negative exponential fashion (Fig. 4c). This indicates more gradual changes in predator abundances prior to large-scale environmental disturbances. The scarcity of historical community-level data within this ecosystem prior to 2000 further limits hypothesis development and testing of the influence of biological parameters on ecosystem patterns, and highlights the need for continued consistent, long-term in situ datasets that cannot be obtained via remote sensing.

Our results indicate a potential return of kelp under a forecasted scenario of mean SST and nitrate conditions, but that a full recovery is suppressed by urchin herbivory (Supplementary Fig. 1). Therefore, it is likely that additional mechanisms beyond a return to mean environmental conditions will be necessary in northern California to reduce urchin population densities to enable a phase shift back to forested conditions. Historically, natural processes such as density-dependent sea urchin disease outbreaks[53] and exposure to large ocean swell events[54] induce mass mortality of urchins. In the absence of urchin disease or effective human intervention to reduce grazer densities, the existing widespread extent of urchin barrens may continue long into the future with devastating impacts to forest-associated fisheries.

We show that this persistent multi-year event has not been seen in the region for the observable past. As a result, innovative management strategies will need to be developed to address the broad-scale collapse of bull kelp forests in northern California and the loss of the fisheries this system once supported. Furthermore, managers of canopy-forming kelp forest ecosystems around the world should work to prioritize time-series measurements of remotely sensed and in situ data for biological and environmental parameters before, and even after, ecosystem shifts occur. Long-term time series can be used to quantify historical baselines, set thresholds for monitoring criteria, develop restoration targets, and track ecosystem recovery. In addition, the implementation of environmental forecasting models[55] should be used to determine if current and future environmental and/or biological conditions are impeding kelp recovery or the likely persistence of recovered

forests. Establishing these adaptive management techniques for perturbed and healthy coastal ecosystems around the globe is crucial for understanding and predicting phase shift dynamics[56,57] and restoring foundation species, and the ecosystem services they provide, especially in the face of increasing frequency and intensity of MHWs as a result of climate change.

## Methods

**Determining kelp canopy coverage.** Bull kelp forests are readily identified by multiple existing high spatial resolution satellite and airborne platforms because their floating surface canopies have strong reflectance in the near-infrared, similar to terrestrial vegetation, and are optically distinct from the surrounding water. We utilized kelp's spectral signature to generate a remote timeseries of bull kelp canopy coverage in order to investigate the influence of environmental factors on canopy area across more than three decades (1985–2019) of US Geological Survey (USGS) Landsat imagery. Cloud-free Landsat 5, 7, and 8 imagery were collected as close to historical maximum canopy extent as possible (August through early-November) and analyzed using multiple endmember spectral mixture analysis (MESMA)[58], which is more robust than band ratio methods such as the Normalized Difference Vegetation Index (NDVI), a commonly used algorithm for detecting kelp[38,59].

**Environmental and biological drivers of bull kelp canopy.** Although most kelp decline occurs at small scales driven by local processes[38], losses of northern California bull kelp in 2014 occurred across nearly 350 km of continuous coastline (Fig. 1b). Therefore, the environmental and biological drivers of kelp were investigated in the context of regional-scale kelp dynamics that occurred across the NE Pacific MHW (Supplementary Table 3 and Supplementary Fig. 3). The large-scale environmental forcings included the Multivariate El Niño/Southern Oscillation Index (MEI), NPGO, and PDO. Local-scale environmental forcings included multiple signatures corresponding with coastal upwelling dynamics, including SST and surface nitrate concentrations ($NO_3$), and significant wave height ($H_s$).

The climatology for each index was removed and then the index was standardized to each variable's mean and standard deviation. For environmental indices (1985–2019) that were measured at hourly ($H_s$) and daily frequency (SST and $NO_3$), data were temporally binned into monthly averages from 1985 to 2019. Standardized indices were calculated by removing the long-term monthly climatology from absolute monthly means and normalizing to the standard deviation. To scale monthly indices to the annual frequency of the kelp index, the monthly climatologically corrected indices were averaged to annual or seasonal values (e.g., spring SST and spring $NO_3$). The annual frequency of MHW days was determined from daily satellite SST measurements based on published methodology from Hobday et al.[60] Sea surface $NO_3$ concentrations were calculated from SST-$NO_3$ relationships developed for northern California by Garcia-Reyes et al.[61] (Supplementary Table 3).

Biological indices, including purple sea urchin (*Strongylocentrotus purpuratus*) and sunflower star (*Pycnopodia helianthoides*) densities, were obtained from California Department of Fish and Wildlife (CDFW; 2003–2018) and Reef Check California (2007–2018)[62] subtidal rocky reef habitat surveys (Supplementary Fig. 4). Both organizations conduct annual surveys through the summer and early fall in northern California. Reef Check California utilizes trained citizen scientists to support coastal ecosystem monitoring, management, and to promote stewardship of sustainable kelp forest communities. Reef Check California surveyed 27 sites in the northern California Sonoma and Mendocino counties between 2007 and 2018 (Supplementary Fig. 4) ranging between 7 and 22 sites annually. Each site consisted of 2 depth strata (inshore: 0–10 m and offshore: 10–20 m) and 6, 60 m² (2 × 30 m) invertebrate and algal transects. Three 60 m² transects were conducted in each depth strata generally parallel to the shore. Partial transects densities were calculated when more than 50 individuals of a species were counted along within a distance of at least 5 m.

CDFW subtidal surveys occur as part of the agency's kelp ecosystem management program[63]. CDFW surveyed 12 sites in the northern California Sonoma and Mendocino counties between 2003 and 2018 (Supplementary Fig. 4) ranging between 2 and 11 sites annually. Random transects were placed within four depth strata (0–4.5 m, 4.5–8.3 m, 9–13.7 m, and 13.7–18.3 m), divided equally within the full depth range (0–18 m) with 60 m² transects (2 × 30 m). Transect locations were at predetermined random GPS coordinates >70 m apart and generally parallel to shore. At each site 15–55 transects were surveyed, with equal numbers of transects per depth stratum. All organisms were counted and recorded within the transect area regardless of density distribution.

Annual densities for the entire study region were determined by taking the mean of all 60 m² transects conducted by Reef Check California and CDFW. Standardized indices were calculated by removing the long-term annual climatology from absolute annual means and normalizing to the standard deviation.

**Statistics and reproducibility: determining the drivers of bull kelp canopy coverage.** Following determination of maximum annual kelp canopy coverage, a PLSR[64] was used to investigate the temporal response of kelp canopy coverage in northern California (Mendocino and Sonoma Counties; Fig. 1a) to large- and local-scale oceanographic and biological processes. PLSR combines principle component analysis (PCA) and multiple linear regression to maximize covariance between the predictor and response variables. This method works particularly well where (1) strong collinearity occurs between predictor variables (Supplementary Fig. 5) and (2) a relatively low number of observations would otherwise reduce model performance[64]. Many of the variables used in this study present strong multi-collinearity (Supplementary Fig. 5) especially between seasonal SST, seasonal $NO_3$ and PDO.

PLSR analysis was conducted using the PLSRegression function in the Python 3.7 sklearn.cross_decomposition machine learning statistical module. Using a k-fold cross-validation technique, environmental variables were selected by calculating the mean squared error (MSE) and determining the optimal configuration via the lowest MSE (Supplementary Fig. 6a). The cross-validation showed that the number of predictor variables had little influence on the performance of the environmental-only indices' first component (Supplementary Fig. 6a). Therefore, a one component, 9 variable configuration was selected. Although a two component, 4 variable configuration was optimal for environmental and biological indices combined (Supplementary Fig. 6b), the study's goal was to compare how the 'environmental-only' model results changed when adding biological forcing (purple urchins). As a result, a one component, 10 variable configuration was selected.

After determining and modeling important drivers of kelp canopy area, an ordinary least-squares regression (OLSR) approach was utilized to understand significant and insignificant temporal changes in relevant biological and environmental indices (kelp canopy area, spring $NO_3$, MHW Days, sunflower star density, and purple urchin density) across the entire timeseries, prior to the NE Pacific MHW (pre-2019), and following the NE Pacific MHW (post-2013). For all indices, except the pre-2014 sunflower star index (which was optimized to a polynomial LSR), OLSRs were fit to data. This simple correlative approach was valuable for understanding how these relevant variables changed on both long and short-term timescales with relevance to dramatic declines in kelp canopy coverage. Trendline error (Supplementary Fig. 2) and regression statistics (Supplementary Table 2; slope, $r^2$, and p-value) are presented in the Supplementary material.

**Reporting summary.** Further information on research design is available in the Nature Research Reporting Summary linked to this article.

## Data availability

Source data for kelp canopy area, large- and local-scale indices were available through public data sources and are listed in detail in Supplementary Table 3. Source data for biological indices are available upon request from Reef Check California and California Department of Fish and Wildlife. All other data that support the findings of this study are available from the corresponding author upon reasonable request.

## Code availability

Code is available on GitHub: https://github.com/mmcph005/PyMESMA.git

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

## Acknowledgements

We thank Reef Check California Program for contribution of subtidal survey data and the many volunteers who devote their time and energy to citizen science efforts. M.H.C. received support from the National Science Foundation (OCE-1538582).

## Author contributions

All authors contributed to this manuscript including conceptualization, M.L.M., R.M.K., and M.H.C.; methodology, M.L.M., R.M.K., D.J.I.F, H.F.H., T.W.B.; writing—original draft preparation, M.L.M.; writing—review and editing, M.L.M., R.M.K., M.H.C., D.J.I.F., H.F.H., T.W.B., L.R.B.

## Competing interests

The authors declare no competing interests.
