## [Peer Review File · Communications Biology]

Reviewers' comments:

Reviewer #1 (Remarks to the Author):

This study used long-term records of kelp forests, environmental variables, and biological factors, to document a shift in a temperate coastal ecosystem. They show that temperature events, namely MHWs drove the substantial loss of kelp forests along N California, following the 'Blob'. Yet, unlike previous events, where kelp recovered, increased sea urchins densities prevented kelps from reestablishing, and the system remained collapsed, with the lowest consecutive measures of kelp abundance over the last 34 years occurring since 2015. The manuscript represents a comprehensive assessment of a climate driven transition on a temperate reef, and explores key mechanisms preventing recovery. The dataset and analyses are well presented, the finding novel, and the results should influence how we understand the stability of ecosystem state shifts. I have some questions regarding the interpretation of some of these findings. Including the coupling of nutrients and temperature, and some questions about the sea urchin data. I also found that the broader consequence of this study, specifically the 'implementation strategies for how this can be used to develop innovative management techniques for restoration and monitoring' are not as well presented as they could be nor supported by the manuscript. Yet these are relatively minor comments. On the whole this is a strong addition to the subject matter and will add much to our current understanding of these dynamics.

Comments by line.

Line 28. I have some questions regarding the statement of temperature-nutrient dynamics as a key extreme event. Although I agree that temperature and nutrients are intrinsically linked in the northern California upwelling system, I am not convinced that they are always linked in coastal ecosystems, and that these variables can be coupled in this way when discussing global trends. In fact, I would say this strong coupling is somewhat unique to eastern boundary regions. Much of the work on extreme events has shown that temperature alone, not nutrients, are the main driver of change. This therefore does not reflect the broader literature and recent syntheses of the ecological impacts of these events. I suggest revising the manuscript to highlight that in this system and SOME others, nutrients are coupled to extreme marine heatwaves. But that extreme temperature alone also drives ecological changes in other regions.

Line 36. I would consider this to be environmental conditions, not physical. Is there a reason for the use of the word physical?

Line 37. Ecological? Not ecosystem... I assume you are referring to the sea urchins and sea stars here?

Line 39. This, in my opinion, is the weakest part of this strong manuscript. It is vague. Replacing it with a sentence describing what the strategy actually is, would be a more useful finding and broader consequence of this study.

Line 42. What does 'ecosystem function and response mean' in this context? Can this really be 'impacted by climate change'. Or is it a consequence?

Line 47. Are you referring to examples of extreme events, or just environmental conditions in general? (drought, heavy rainfall, and cold)... As the sentence is now, it reads as extreme events, but would be nice to be more explicit. Also, what about including heatwaves here? And are extreme cold events more common in terrestrial systems than warm ones, especially with climate change?

Line 50. This understanding that 'storms and temperature-nutrient dynamics generally drive change in coastal marine systems' is only referenced by two regional papers from this study area. I argue that the drivers of change in the coastal zone, globally, are more complex. Bleaching on the Great Barrier Reef from MHWs, for example, is not tied to nutrients. The statement is also missing a suite of human impacts, such as coastal development, deforestation, pollution. Perhaps stating that these are examples of environmental drivers, not general drivers, would be helpful. See recent paper by Holbrook et al. 2019 and Smale et al. 2020 about impacts of MHWs.

Smale, D.A., et al. Marine heatwaves threaten global biodiversity and the provision of ecosystem services. *Nat. Clim. Chang.* 9, 306–312 (2019). <https://doi.org/10.1038/s41558-019-0412-1>.
Holbrook, N.J., Scannell, H.A., Sen Gupta, A. et al. A global assessment of marine heatwaves and their drivers. *Nat Commun* 10, 2624 (2019). <https://doi.org/10.1038/s41467-019-10206-z>

Line 95. Here the use of environmental seems more appropriate than the previous use of physical variables. Suggest changing throughout.

Line 120. Yellow.

Line 131. It would be useful to have a line here or in the methods about model selection, and why this approach may deal with lower sample sizes and high numbers of predictor variables.

Line 142. Is this marine heatwave days? I do not see heatwave years in fig3, perhaps need a reference to the supplement.

Line 182. Not sure that period is the word here. Fluctuating, or variable perhaps?

Line 184. Here it would be useful what changes in biological indices were observed in S California and N Baja, unclear if this suggests there was no stepwise change in sea urchins or urchin predators or kelp or all three.

Line 192. What is an endmember, is this referring to the northern or southern limits in fig 1?

Line 198. Resiliency is not a word. Change to resilience.

Line 203-205. This section is confusing. It seems to suggest that predators do not target barrens urchins because they have poor condition. In fact, there are a number of examples of predators eating urchins in barrens and restoring the system back to a kelp forest. Sea otters in Alaska and British Columbia, king crabs in Norway, cancer grabs in Gulf of Maine, and predator fish and lobster in New Zealand, to name a few. Urchin condition can be important for urchin fisheries if they are keeping populations in control.

Line 207. In the past periods of low kelp abundance in figure 2, are there any reports of sea urchins also increasing (1995, 1997-98). It is curious that in 2006, the purple urchin did not change densities, despite the loss of much of the kelp canopy. What was the system state during this time? Was it no canopy, but still understory, subsurface macroalgae? The ecology of these dynamics would be interesting to explore.

Lines 211. I do not really see the link that lack of temperature dependence and why recovery is elusive. Is it because temperatures have recovered so this should allow the population to rebound if they were thermal limited?

Line 225. Numerical response is an odd way to say predator increase?

Line 225. Does this mean intervention to increase sea stars or reduce urchins?

Line 231. How will prioritizing time series measures help address this collapse? Wouldn't that take another decade? Is it not too late?

Line 237. Seems like understanding the population dynamics of sea urchins in this region would be a critical line of research given these findings.

Line 263. What region were the sea urchin and sunflower star data collected for from the CDFW and Reef check data. How does this scale compare to the scale of the satellite observations?

Supplementary material 3.

What do the bolded cells represent? Is this appropriate to do for MHW days, as these are discrete events?

Discussion. What about a comparison to the findings from Southern California on kelp forest loss with extreme MHWs? I would be curious to see how these results fit in with dynamics in that nearby area. Are there sea urchins there?

Arafeh-Dalmau N, Montaña-Moctezuma G, Martínez JA, Beas-Luna R, Schoeman DS and Torres-Moye G (2019) Extreme Marine Heatwaves Alter Kelp Forest Community Near Its Equatorward Distribution Limit. *Front. Mar. Sci.* 6:499. doi: 10.3389/fmars.2019.00499

Cavanaugh KC, Reed DC, Bell TW, Castorani MCN and Beas-Luna R (2019) Spatial Variability in the Resistance and Resilience of Giant Kelp in Southern and Baja California to a Multiyear Heatwave. *Front. Mar. Sci.* 6:413. doi: 10.3389/fmars.2019.00413

Reviewer #2 (Remarks to the Author):

This manuscript describes the analysis of a 34 year data set on bull kelp canopy density derived from satellite imagery and explores how kelp density is influenced by a range of physical (e.g. marine heatwaves, nutrients, climate oscillations, wave height) and biological (urchin and sea star numbers) processes. While the data set is comprehensive in terms of space and time, particularly for kelp density the approach taken is purely correlative. There is nothing inherently wrong with this, but throughout I felt that very bold cause and affect statements were made when the evidence for cause and effect did not exist (see below for specifics). I would like to see a more nuanced assessment of the quality of the data and the inferences that can be made based on it. I also found that many general statements were really only applicable to certain systems rather than all kelp ecosystems globally. While the analysis seems appropriate I believe that the language used within the manuscript needs better reflect the inferences that can be made based on the data available before this manuscript should be considered for publication.

Specific comments

Title – the title is an example of the tone of the language. It attributes cause and effect when this is

not possible based on the data available. I suggest the authors alter the title to reflect the type of data they have i.e. there are relationships between oceanographic conditions and ecological perturbations.

Line 36 Delete second but

Line 49 Temperature-nutrient dynamics do not structure marine ecosystems globally. The interaction between temperature and nutrients is really only apparent in upwelling systems, which only occur in a limited number of coastal zones. I suggest the authors rephrase.

Line 51 Marine heatwaves again do not always modify temperature-nutrient dynamics in all locations. Indeed the Hobday et al definition of marine heatwaves is solely based on changes in temperature. I therefore suggest the authors rephrase this sentence.

Line 52 Suggest adding Smale et al 2019 as this paper looks at the impacts of MHWs across coastal systems around the world.

Line 55 Kelp are not just found in upwelling systems. Substantial kelp populations are found in the north Atlantic, southern Australia, temperate east Africa etc – none of these areas are upwelling regions. I therefore suggest the authors rephrase this sentence.

Line 57 Suggest replacing geographic with regional

Line 58 Delete drivers

Line 59 Delete temperate and arctic kelp

Line 61 Should be Western Australia as it is the name of an Australia state.

Line 72-72 A reference is required to support the statement that sunflower stars are now the key predator of urchins

Line 75-78 An evidence base is required to support the statement that kelp persistence was balanced by top-down urchin predation by sunflower stars

Line 84 The authors need to make it clear that such satellite approaches are only possible for canopy forming kelps such as bull and giant kelp. This technique cannot be used for stipitate or prostrate kelp species.

Line 91-94 I find this a very strong cause and effect statement given the data to hand. I suggest the authors modify the strength of the language to reflect the type of data and analysis that they have undertaken.

Line 106 Consider adding “even though environmental conditions become more favourable to kelp” after 2019.

Line 109 It would be useful to have Punta Gorda indicated on the map for those not familiar with the area. While not knowing where Punta Gorda is it would appear that kelp became patchy in this northern region well before 2008, indeed before 1990. Perhaps Punta Gorda is closer to Fort Bragg, but it might be worth the authors discussing this loss of kelp in the far north as this doesn't seem aligned to the perturbations that are used to explain the decline in later years.

Line 130 Delete and Hs as comes later in the sentence

Line 139-140 This is a pretty big statement to make when a simple correlative approach has been taken. How can the authors be sure that change in sea urchin numbers is not driven by changing physical conditions.

Line 147-148 I don't know how the reader is supposed to interpret this. The figure merely shows that when urchins are in the model kelp abundance stays low. It doesn't necessarily show that a barren has formed. I suggest restructuring the sentence.

Line 174-178 Without experimental evidence or reference to the literature this is all conjecture, but is written as if there is evidence of cause and effect. As with other sections I think this needs to be toned down.

Line 183-187 I don't really understand what message the authors are trying to convey here. I am not clear on what the biological indices were in the MHW. I am also not clear about how giant kelp differed from bull kelp in their responses. I suggest the authors clarify their language.

Line 192 I am not clear what the authors mean by endmembers

Line 202-204 An evidence base to support this statement is required

Line 211-212 I am not sure how the previous sentence leads the reader to this conclusion. I suggest the authors make this connection more clearly.

Fig 2d Given the amount of temporal site to site variation in kelp density I am not sure it is appropriate to amalgamate across the whole study region. Or perhaps the authors just need to justify this approach in the methods. In the figure legend the authors need to state that orange is where the blob and the el nino even overlap.

Fig 3 From the methods it would appear that the authors have not checked for collinearity between physical predictors. It is likely that many of these will be correlated and therefore it may not be appropriate to have them all in the model. The acronyms need to be defined in the figure legend.

Fig 4 Add P to LSR in the figure legend

S3 The authors should state in the legend that the grey cells represent significant relationships.

Reviewer #3 (Remarks to the Author):

Review of 'Oceanographic and ecological perturbations associated with a marine heatwave dramatically shift the structure of a temperate coastal marine ecosystem' by McPherson et al
This paper describes the collapse and conversion of large bull kelp forests, and decreases in sea star populations in Northern California to urchin dominated barrens.

The paper is well written, and the findings are clearly important to show.

My main concern is (probably not surprising) that two of the Authors recently told the same general story in Scientific Report (2019, 9:15050) and that the novelty therefore is minimal. Indeed, I see no new ecological take home messages when comparing the abstracts between the 2019 scientific report paper and the present manuscript. I am, of course, aware that there are methodological differences between the papers – and that the time series data are longer for this new paper – but that does very

little to change the overall impression about the findings.

So the fate of this manuscript very much lies with the editor – i.e., how 'novel' should the results be before a paper can be published in this particular journal.

Below I outline a few minor comments in chronological order.

L32-34. It sounds like the satellite images can be used to detect declines in sea stars (which they clearly cannot – only kelp cover)

L72-73. Are there no other natural urchin predators than otter? No fish that will eat the urchin?

L85. You say that satellite images are great to monitor kelp – however, its only really useful for kelp with surface canopies, i.e. *Macrocystis* and *Nereocystis* – most kelp species cannot be detected with satellites.

L90 – this again sounds like the satellite images can be used to detect declines in sea stars (which they clearly cannot – only kelp cover)

L94-96. How does your work provide 'context for monitoring'? - as I said, most kelp systems cannot be monitored with satellites.

Somewhere in the introduction: I am missing a bit of natural history of *Nereocystis*. In particular that it only lives to 12-18 month (which is pretty unusual for a very large kelp) – so that there are very large phenological short term and small scale fluctuations in canopy cover.

L145-147. This could be a place where you discuss problems associated with low propagule pressure (as another reason why recovery is slow). You spend a lot of time discussing urchins and top-down control as a future limitations. I think you should include more discussion on how few propagules are now being released to the system – and that, by itself, limit recovery.

L155. The only p-value in the entire paper. When I compare to the method section it is not clear what test you used, what assumptions was required or why you did this particular test – and why you did not do more tests)

L166. Here is an example on means and slope values – but without p-values – so why is this a different approach compared to L155?

L183-184. Its interesting that *Macrocystis* was less affected by urchins/mhw/ssws. It would be nice to know how Macro and Nereo have different temperature sensitivities and what there latitudinal ranges are.

L202-205. I would question that lower nutritional values in urchins limits recovery of predators. From your photos it looks in scientific report it looks like there are so many urchins that are easy to pick for predators that a slightly smaller nutritional value is off-set by a much lowered searching/catching/handling time.

L224-227. Again, are there no other predators than otters? Here it is related to management – is otter really the only 'natural' option for urchin control?

L261-263. Here you make a simple reference to another paper that describe the sampling methods related to sea stars and urchins. However, these biological data are really important so the reader should know much more about the methods; how many samples, where, when and at what depth? Without such standard information readers cannot evaluate the strength the trends.

Somewhere in the method section. I would like to see some clear descriptions of what statistical tests you used, why and how test assumptions are addressed.

I would also like to see some sort of variability associated with some of the graphs. Currently no data variability is shown so I have no idea about how robust your data are. I would also like to know about sample sizes in the figure legend (scientific figures + legends should preferably show some kind of central values, some kind of data dispersion and the associated replication levels)

I also think the ecological context and discussion could benefit from invoking trophic cascades and alternative stable state concepts/theories (which are the dominant ecological processes going on here – but none of these classic ecological concepts are mentioned in the paper).

Referee expertise:

Referee #1: Kelp ecology

Referee #2: Marine ecology, marine heatwaves

Referee #3: Marine ecology, marine heatwaves

Reviewers' comments:

Reviewer #1 (Remarks to the Author):

This study used long-term records of kelp forests, environmental variables, and biological factors, to document a shift in a temperate coastal ecosystem. They show that temperature events, namely MHWs drove the substantial loss of kelp forests along N California, following the 'Blob'. Yet, unlike previous events, where kelp recovered, increased sea urchin densities prevented kelps from reestablishing, and the system remained collapsed, with the lowest consecutive measures of kelp abundance over the last 34 years occurring since 2015. The manuscript represents a comprehensive assessment of a climate driven transition on a temperate reef, and explores key mechanisms preventing recovery. The dataset and analyses are well presented, the finding novel, and the results should influence how we understand the stability of ecosystem state shifts. I have some questions regarding the interpretation of some of these findings. Including the coupling of nutrients and temperature, and some questions about the sea urchin data. I also found that the broader consequence of this study, specifically the 'implementation strategies for how this can be used to develop innovative management techniques for restoration and monitoring' are not as well presented as they could be nor supported by the manuscript. Yet these are relatively minor comments. On the whole this is a strong addition to the subject matter and will add much to our current understanding of these dynamics.

We thank the reviewer for their praise and support of the manuscript and feel that their comments and suggestions have greatly improved the quality of the manuscript. In response, we have specifically (1) edited language regarding temperature-nutrient dynamics and clarified these dynamics with respect to eastern boundary current systems (in the abstract and entire first paragraph of the introduction), (2) clarified several aspects of our discussion with regard to urchin recruitment dynamics in northern California and a nuanced discussion of the role of predators in urchin barrens (of which the reviewer expressed confusion to statements made in the original manuscript), and (3) developed a more robust link between the presentation of the development of management techniques and restoration strategies in the introduction and discussion.

Comments by line.

Line 28. I have some questions regarding the statement of temperature-nutrient

dynamics as a key extreme event. Although I agree that temperature and nutrients are intrinsically linked in the northern California upwelling system, I am not convinced that they are always linked in coastal ecosystems, and that these variables can be coupled in this way when discussing global trends. In fact, I would say this strong coupling is somewhat unique to eastern boundary regions. Much of the work on extreme events has shown that temperature alone, not nutrients, are the main driver of change. This therefore does not reflect the broader literature and recent syntheses of the ecological impacts of these events. I suggest revising the manuscript to highlight that in this system and SOME others, nutrients are coupled to extreme marine heatwaves. But that extreme temperature alone also drives ecological changes in other regions.

We agree that we were not specific enough in distinguishing the differences in global and regional dynamics of temperature-nutrient dynamics in coastal marine ecosystems and thank the reviewer for their comment. As a result, we have rephrased the beginning of the abstract to emphasize that MHWs are a global phenomenon but specifically influence EBC systems temperature-nutrient dynamics.

L 26 – 28: “Climate change is responsible for increased frequency, intensity, and duration of extreme events, such as marine heatwaves (MHWs). Within eastern boundary current systems, MHWs have profound impacts on temperature-nutrient dynamics that drive primary productivity.”

Furthermore, we have re-written the first paragraph of the introduction to reflect the reviewer’s comment and suggestions by specifically addressing drivers of change in coastal marine ecosystems. For further detail, see our response to the reviewer’s comments for L 42, L 47, and L50.

Line 36. I would consider this to be environmental conditions, not physical. Is there a reason for the use of the word physical?

Thank you for your question. The only reason we use the word ‘physical’ rather than ‘environmental’ is to specify the variables used in this analysis linked to physical drivers of oceanographic trends. However, we agree that the word ‘environmental’ may be more intuitive for readers outside of a specific oceanographic mindset. Therefore, we have changed the use of the word ‘physical’ to ‘environmental’ throughout the manuscript.

Line 37. Ecological? Not ecosystem... I assume you are referring to the sea urchins and sea stars here?

L 36: We have changed ‘ecosystems’ to ‘biological’ to reference specific organisms rather than the entire ecosystem and remain consistent with language used throughout the rest of the manuscript.

Line 39. This, in my opinion, is the weakest part of this strong manuscript. It is vague. Replacing it with a sentence describing what the strategy actually is, would be a more useful finding and broader consequence of this study.

We agree that the statement was vague and weakly described the management strategies that we later describe in more detail in the discussion. We have worked to clarify how this important timeseries can be used for management strategies in the abstract. This coincides more appropriately with text in the introduction and discussion.

L 38 – 42: Based on our findings, we recommend the implementation of ecosystem-based and adaptive management strategies, such as (1) monitoring the status of key ecosystem attributes: kelp distribution and abundance, and densities of sea urchins and their predators, (2) developing management responses

to threshold levels of these attributes, and (3) creating quantitative restoration suitability indices for informing kelp restoration efforts.”

Line 42. What does ‘ecosystem function and response mean’ in this context? Can this really be ‘impacted by climate change’. Or is it a consequence?

In order to address the next 3 very astute comments and questions regarding the first paragraph of the manuscript (previously L 42, L 47, and L 50), we have edited the entire first paragraph to more specifically address drivers of change in coastal marine ecosystems, including clarifying the impacts of climate change globally and within eastern boundary systems:

L 45 - 48: Coastal marine ecosystem response to climate change in the 21st century is predicted to manifest in various ways, including through habitat contraction, species range shifts, and losses of biodiversity and functionality¹. These responses can manifest through both long-term gradual changes and more episodic events^{2,3}.

Additionally, we have included a brief comparison of the differences between drivers of MHWs in coastal systems globally versus eastern boundary currents. These changes help clarify the variability across marine systems rather than generalizing all terrestrial systems and all marine systems.

L 48 - 62: However, it can be difficult to distinguish the impacts of gradual (e.g., increasing mean temperatures) and irregular (e.g., increasing storm frequency) climate-induced shifts from changes in underlying naturally stochastic events (e.g., El Niño Southern Oscillation (ENSO)). One such example is the ocean warming phenomenon of marine heatwaves (MHWs). Global oceanic and atmospheric drivers influence regional frequency, duration, and intensity of MHWs^{4,5}, all of which are increasing^{4,6}. In eastern boundary current ecosystems, such as the California Current, MHWs are highly correlated to changes in nutrient availability given the strong correlation between temperature and nutrients^{7,8} (e.g., anomalously high sea surface temperature (SST), low nitrate concentration [NO₃]). MHWs can have notable impacts on coastal ecosystems, such as seagrass beds⁹, coral reefs¹⁰ and kelp forests¹¹, and especially on the foundation species and ecosystem engineers (e.g., seagrasses, corals, kelps) that define these systems. Furthermore, the specific impacts of climate induced changes to these habitat forming sessile organisms via the coupled impacts of regional non-climate change human influences and species thermal tolerance, greatly increases their vulnerability relative to mobile species¹².

Line 47. Are you referring to examples of extreme events, or just environmental conditions in general? (drought, heavy rainfall, and cold)... As the sentence is now, it reads as extreme events, but would be nice to be more explicit. Also, what about including heatwaves here? And are extreme cold events more common in terrestrial systems that warm ones, especially with climate change?

We removed the discussion of terrestrial drivers completely to focus on coastal marine ecosystems. We have also emphasized MHWs earlier in the paragraph to focus the introduction. Please see the above comment for our full changes to this section of the manuscript.

Line 50. This understanding that ‘storms and temperature-nutrient dynamics generally drive change in coastal marine systems’ is only referenced by two regional papers from this study area. I argue that the drivers of change in the coastal zone, globally, are more complex. Bleaching on the Great Barrier Reef from MHWs, for example, is not tied to nutrients. The statement is also missing a suite of human impacts, such as coastal

development, deforestation, pollution. Perhaps stating that these are examples of environmental drivers, not general drivers, would be helpful. See recent paper by Holbrook et al. 2019 and Smale et al. 2020 about impacts of MHWs.

Smale, D.A., et al. Marine heatwaves threaten global biodiversity and the provision of ecosystem services. *Nat. Clim. Chang.* 9, 306–312 (2019). <https://doi.org/10.1038/s41558-019-0412-1>.
Holbrook, N.J., Scannell, H.A., Sen Gupta, A. et al. A global assessment of marine heatwaves and their drivers. *Nat Commun* 10, 2624 (2019). <https://doi.org/10.1038/s41467-019-10206-z>

Please see the above 2 comments to address these concerns. More specific discussion of global drivers of MHWs referencing Smale et al. 2019 and Holbrook et al. 2019 (reference numbers 12 and 5, respectively) is included as part of these edits.

Line 95. Here the use of environmental seems more appropriate than the previous use of physical variables. Suggest changing throughout.

Changed based on comment from L 36 above and justified in the above author response.

Line 120. Yellow.

Thank you, this has been corrected (L 151).

Line 131. It would be useful to have a line here or in the methods about model selection, and why this approach may deal with lower sample sizes and high numbers of predictor variables.

We thank the reviewer for the suggestion to include more details on our model selection and have included more language into the Methods:

*L 384 - 389: "PLSR combines principle component analysis (PCA) and multiple linear regression to maximize covariance between the predictor and response variables. This method works particularly well where (1) strong collinearity occurs between predictor variables (**S6**) and (2) a relatively low number of observations would otherwise reduce model performance¹³. Many of the variables used in this study present strong multi-collinearity (**S6**) especially between seasonal SST, seasonal NO₃ and PDO."*

Furthermore, the selection of PLSR is warranted and validated by Carrascal et al. (2009) (reference number 66), who carried out a simulation experiment to compare PLSR with multiple regression (a type of GLM) and with a combination of PCA and multiple regression. They varied the number of predictor variables and sample sizes. Although the PLSR models explained a similar amount of variance to those results obtained by the other techniques, they were more reliable than other techniques when identifying relevant variables and their magnitudes of influence, especially in cases of small sample size and low tolerance.

Line 142. Is this marine heatwave days? I do not see heatwave years in fig3, perhaps need a reference to the supplement.

In this sentence we are referring to model and forecast results based on PLSR runs from the Figure 3a configuration. By altering the model years (black solid line in Fig. 3c) between 1985 – 2013 and 1985 – 2016, we were able to compare forecasted results with the NE Pacific MHW years included and excluded.

From this comparison we determined that the forecasted model responded similarly whether the NE Pacific MHW was included or not. We have attempted to clarify this sentence in the manuscript to more clearly represent the approach taken.

L 175 - 178: "Environmental drivers correctly represented bull kelp response to low NO₃ and high SST conditions across the NE Pacific MHW regardless of whether the event was included in the temporal representation of the PLSR model (Fig. 3c solid black line) and forecast (Fig. 3c dashed black line) results."

Line 182. Not sure that period is the word here. Fluctuating, or variable perhaps?

L 220: We have changed 'periodic' to 'fluctuating' for clarity.

Line 184. Here it would be useful what changes in biological indices were observed in S California and N Baja, unclear if this suggests there was no stepwise change in sea urchins or urchin predators or kelp or all three.

We were referring solely to the stepwise changes in kelp and not other biological indices. Recognizing that the sentence was misleading and confusing, we have attempted to clarify the language.

*L 224-227: "This is evidenced by the fact that the stepwise decline in northern California bull kelp canopy area across the NE Pacific MHW was not observed in giant kelp (*Macrocystis pyrifera*) canopy biomass at a regional scale in southern California¹⁹ and northern Baja California^{19,20,24}."*

Line 192. What is an endmember, is this referring to the northern or southern limits in fig 1?

Yes, precisely. However, to help reduce confusion by future readers we have changed the wording and moved this to a more appropriate section of the paper earlier in the results within the description of figure 1.

L 136-139: "The northern and southern portions of the historical kelp canopy distribution within our study area are regions characterized by sandy sediment⁴⁰ (poor substrate for kelp spore settlement³⁶), resulting in sparser and patchier distribution than the rockier coastline between Fort Bragg and Jenner prior to the NE Pacific MHW."

Line 198. Resiliency is not a word. Change to resilience.

According to Merriam Webster Dictionary, resiliency is a word and interchangeable with the word resilience but used less commonly. Seeing as this is a minor comment, we chose to keep the use of the word throughout the manuscript.

Line 203-205. This section is confusing. It seems to suggest that predators do not target barrens urchins because they have poor condition. In fact, there are a number of examples of predators eating urchins in barrens and restoring the system back to a kelp forest. Sea otters in Alaska and British Columbia, king crabs in Norway, cancer grabs in Gulf of Maine, and predator fish and lobster in New Zealand, to name a few. Urchin condition can be important for urchin fisheries if they are keeping populations in control.

We have expanded our discussion of why urchin barren conditions can experience diminished predation and are harmful to ecosystem recovery, especially, as in this case, a near complete absence of urchin

predators and included four additional references to studies demonstrating this (Smith et al. 2020, Babcock et al. 2010, Eurich et al. 2014, and Graham 2004).

L 262 - 272: "In sea urchin barrens, urchins are starved and lack energetic value to predators with high metabolic rates⁵⁰. Moreover, behaviorally mediated predators often track changes in the distribution of profitable prey, which further complicates implications for recovery. Urchin barrens are characterized by low urchin gonad index; because the gonads are what urchin predators target and consume, urchin barren states potentially limit increased predation of this particular prey⁵¹. The high urchin densities observed in northern California have induced starvation conditions and reduced the nutritional value of the urchins⁵¹. Additionally, since the overall ecosystem biodiversity of urchin barrens is severely reduced⁵², opportunities for predator (otters, sunflower stars, etc.) recovery is diminished. Ecosystem recovery is further limited by evidence that the effects on prey can lag behind the recovery of a predator⁵³."

Line 207. In the past periods of low kelp abundance in figure 2, are there any reports of sea urchins also increasing (1995, 1997-98). It is curious that in 2006, the purple urchin did not change densities, despite the loss of much of the kelp canopy. What was the system state during this time? Was it no canopy, but still understory, subsurface macroalgae? The ecology of these dynamics would be interesting to explore.

We feel that the reviewer has brought up a very important point regarding urchin dynamics and have included a full paragraph in the discussion addressing these details.

L 240-255 : "Regional-scale sea urchin larval recruitment dynamics are associated with large-scale environmental drivers and subsequent population dynamics⁴³. Anomalously high larval recruitment was observed in Fort Bragg, CA, peaking in 2015 and increased larval settlement appeared to be correlated with juvenile and adult urchin densities during the NE Pacific MHW. Although there are no reliable in situ data available for sea urchin densities prior to 2003 from the northern California region, there are purple urchin settlement data as early as 1990 from Fort Bragg (1990-2016⁴³) and Westport, Pt. Cabrillo, and Pt. Arena (1990-1993⁴⁷). These settlement records show there was anomalously high larval settlement from 1993 to 1994 and 1998. Despite bull kelp canopy area being anomalously low between 1995 and 1998 (Fig. 4b), there is no evidence of complete kelp forest collapse or ecosystem shift during that period and no way to verify that high juvenile urchin densities coincided with the high larval densities. Furthermore, anomalously low kelp conditions do not always follow or co-occur with sea urchin larval settlement events (e.g. Fig. 4b 2003 – 2007; Okamoto et al. 2020 Fig. 3a). Given the positive relationship between SST and larval settlement in northern California⁴³ and predictions of more frequent and/or severe MHW⁴⁸, restoration efforts in northern California would benefit from a greater understanding of localized sea urchin population dynamics in that region."

Lines 211. I do not really see the link that lack of temperature dependence and why recovery is elusive. Is it because temperatures have recovered so this should allow the population to rebound if they were thermal limited?

We understand that the sentence structure may have caused some confusion and we have reworded it for clarity.

L 286 - 294 : "Evidence suggests that the pathogen associated with SSWS in the sunflower star was not temperature dependent, nor responsible for disease observed in other asteroids throughout the region⁵⁴. This may explain why recovery for sunflower stars across the region, and in turn kelp forest recovery in northern California, remain absent despite temperature and nutrient conditions recovering slightly in 2017.

Furthermore, the clear phase shift observed in other biological and environmental conditions in northern California (Fig. 4b, d, and e), such as sunflower star populations, began to decline well before the NE Pacific MHW in a negative exponential fashion (Fig. 4c). This indicates more gradual changes in predator abundances prior to large-scale environmental disturbances.”

Line 225. Numerical response is an odd way to say predator increase?

Thank you for the suggestion, we have edited this section for clarity to address the next two comments:

L 303 - 307: “Historically, natural processes such as density-dependent sea urchin disease outbreaks⁵⁵ and exposure to large ocean swell events⁵⁶ induce mass mortality of urchins. In the absence of urchin disease or effective human intervention to reduce grazer densities, the existing widespread extent of urchin barrens may continue long into the future with devastating impacts to forest-associated fisheries.

Line 225. Does this mean intervention to increase sea stars or reduce urchins?

Please see response directly above.

Line 231. How will prioritizing time series measures help address this collapse? Wouldn't that take another decade? Is it not too late?

We believe that monitoring of kelp forests using remote sensing can be highly beneficial, even after an ecosystem shift occurs, especially for setting restoration targets and tracking ecosystem recovery. Since the statement was meant to apply to management of kelp systems globally, we have also clarified the language used.

Rather than specifically implying management in northern California, the sentence has now been broadened to include implications for kelp forests systems globally and specific uses for remote sensing products after and before ecosystem decline.

L 311 - 318: Furthermore, managers of canopy kelp forest ecosystems around the world should work to prioritize time series measurements of remotely sensed and in situ data for biological and environmental parameters before, and even after, ecosystem shifts occur. Long-term time series can be used to quantify historical baselines, set thresholds for monitoring criteria, develop restoration targets, and track ecosystem recovery. Additionally, the implementation of environmental forecasting models²⁷ should be used to determine if current and future environmental and/or biological conditions are impeding kelp recovery or the likely persistence of recovered forests.”

Line 237. Seems like understanding the population dynamics of sea urchins in this region would be a critical line of research given these findings.

We agree that sea urchin dynamics are an important piece of understanding the complex dynamics of northern California kelp forests and the response to the NE Pacific MHW. We have included a sentence at the end of the paragraph devoted to discussing what is known about urchin dynamics in our study region.

L 252-255: “Given the positive relationship between SST and larval settlement in northern California⁴³ and predictions of more frequent and/or severe MHW⁴⁸, restoration efforts in northern California would benefit from a greater understanding of localized sea urchin population dynamics in that region.”

Line 263. What region were the sea urchin and sunflower star data collected for from

the CDFW and Reef check data. How does this scale compare to the scale of the satellite observations?

We utilized two sources for our subtidal biological data (Reef Check California and California Department of Fish and Wildlife) and added a supporting figure presented in the supplemental material (S7). This figure visualizes the temporal and spatial distribution of site coverage and transect area. Transect area is visualized temporally (S7a) and spatially (S7b and c) by year and site, respectively, for both data sources. Panel S7c shows the distribution of sites within the study area. Although there are portions of the coast where survey sites are clustered and areas where survey sites are notably absent, site selection is based on several factors including accessibility, kelp presence/absence, substrate type, and locations of historical abalone recreational fishing grounds. Between the two organizations, nearly 160,000 m² of benthos was surveyed between 2003 and 2018.

A detailed description of the methodology used by each organization to help the reader understand the scale of the measurements has been added to the Methods section as follows:

L 355 - 375: “Biological indices, including purple sea urchin (*Strongylocentrotus purpuratus*) and sunflower star (*Pycnopodia helianthoides*) densities, were obtained from California Department of Fish and Wildlife (CDFW; 2003 - 2018) and Reef Check California (2007 – 2018)⁶⁴ subtidal rocky reef habitat surveys. Both organizations conduct annual surveys through the summer and early fall in northern California. Reef Check California utilizes trained citizen scientists to support coastal ecosystem monitoring, management, and to promote stewardship of sustainable kelp forest communities. Reef Check California surveyed 27 sites in the northern California Sonoma and Mendocino counties between 2007 and 2018 (S7) ranging between 7 and 22 sites annually. Each site consisted of 2 depth strata (inshore: 0 – 10 m and offshore: 10 – 20 m) and 6, 60 m² (2 x 30 m) invertebrate and algal transects. Three 60 m² transects were conducted in each depth strata generally parallel to shore. Partial transect densities were calculated when more than 50 individuals of a species were counted along a distance of at least 5 m.

CDFW subtidal surveys occur as part of the agency’s kelp ecosystem management program⁶⁵. CDFW surveyed 12 sites in the northern California Sonoma and Mendocino counties between 2003 and 2018 (S7) ranging between 2 and 11 sites annually. Random transects were placed within four depth strata (0 – 4.5 m, 4.5 m – 8.3 m, 9 – 13.7 m, and 13.7 – 18.3 m), divided equally within the full depth range (0 – 18 m) with 60 m² transects (2 x 30 m). Transect locations were at predetermined random GPS coordinates greater than 70 m apart and generally parallel to shore. At each site 15–55 transects were surveyed, with equal numbers of transects per depth stratum. All organisms were counted and recorded within the transect area regardless of density distribution.”

Supplementary material 3.

What do the bolded cells represent? Is this appropriate to do for MHW days, as these are discrete events?

Bolded cells represent significant temporal changes in environmental and biological trends associated with Figure 4 indices. Just like other SST indices, the ‘MHW days’ index represents the number of annual MHW days depicted in standard deviations from the long-term mean. Therefore, we can investigate temporal changes in the number of MHW days the same way we investigate changes in other environmental and biological variables and feel that it is appropriate. We have attempted to emphasize the fact that we are presenting temporal trends of the indices rather than discrete events in the S4 table columns and table caption.

Discussion. What about a comparison to the findings from Southern California on kelp forest loss with extreme MHWs? I would be curious to see how these results fit in with

dynamics in that nearby area. Are there sea urchins there?

Arafeh-Dalmau N, Montaña-Moctezuma G, Martinez JA, Beas-Luna R, Schoeman DS and Torres-Moye G (2019) Extreme Marine Heatwaves Alter Kelp Forest Community Near Its Equatorward Distribution Limit. *Front. Mar. Sci.* 6:499. doi: 10.3389/fmars.2019.00499

Cavanaugh KC, Reed DC, Bell TW, Castorani MCN and Beas-Luna R (2019) Spatial Variability in the Resistance and Resilience of Giant Kelp in Southern and Baja California to a Multiyear Heatwave. *Front. Mar. Sci.* 6:413. doi: 10.3389/fmars.2019.00413

*A discussion of these dynamics was presented in the originally submitted manuscript, though we focus on the physiological temperature limitations of the two kelp genera as drivers of their response (now L 219 – 235). With regard to urchin dynamics at the southern range limit of *Macrocystis* we have included the following statement to the end of the paragraph.*

L 235-239: “Furthermore, sea urchin dynamics differed between northern California and southern/Baja California. Increases in sea urchin density (and decreases in invertebrate species richness) in localized areas of the Baja region indicate that grazing pressure, in addition to temperature stress, may have occurred, but not across the entire region.”

This also provided the opportunity to segue into a discussion of localized urchin dynamics in northern California in the following paragraph (L 240-255).

Reviewer #2 (Remarks to the Author):

This manuscript describes the analysis of a 34 year data set on bull kelp canopy density derived from satellite imagery and explores how kelp density is influenced by a range of physical (e.g. marine heatwaves, nutrients, climate oscillations, wave height) and biological (urchin and sea star numbers) processes. While the data set is comprehensive in terms of space and time, particularly for kelp density the approach taken is purely correlative. There is nothing inherently wrong with this, but throughout I felt that very bold cause and effect statements were made when the evidence for cause and effect did not exist (see below for specifics). I would like to see a more nuanced assessment of the quality of the data and the inferences that can be made based on it. I also found that many general statements were really only applicable to certain systems rather than all kelp ecosystems globally. While the analysis seems appropriate I believe that the language used within the manuscript needs better reflect the inferences that can be made based on the data available before this manuscript should be considered for publication.

We thank the reviewer for their candid comments and suggestions and have removed or reworded statements that the reviewer felt were beyond the scope of the analysis and results. Additionally, we have narrowed the introduction and discussion to focus on eastern boundary current systems. Finally,

much detail regarding data acquisition, availability, and quality was added to the Methods and Supplemental Material. We believe that, in combination, the suggestions have greatly improved the quality of the manuscript.

Specific comments

Title – the title is an example of the tone of the language. It attributes cause and effect when this is not possible based on the data available. I suggest the authors alter the title to reflect the type of data they have i.e. there are relationships between oceanographic conditions and ecological perturbations.

Thank you for your suggestion to alter the title to reflect the type of data we have presented. We have selected the following as we feel it describes the main findings of the study without over extrapolating them beyond the correlative approach taken here.

L 1: Large-scale shift in the structure of a kelp forest ecosystem co-occurs with an epizootic and marine heatwave

Line 36 Delete second but

This has been deleted.

Line 49 Temperature-nutrient dynamics do not structure marine ecosystems globally. The interaction between temperature and nutrients is really only apparent in upwelling systems, which only occur in a limited number of coastal zones. I suggest the authors rephrase.

Line 51 Marine heatwaves again do not always modify temperature-nutrient dynamics in all locations. Indeed the Hobday et al definition of marine heatwaves is solely based on changes in temperature. I therefore suggest the authors rephrase this sentence.

Line 52 Suggest adding Smale et al 2019 as this paper looks at the impacts of MHWs across coastal systems around the world.

To address the above three comments for L49, L51, and L52 we have included a brief comparison of the differences between drivers of MHWs in coastal systems globally versus eastern boundary currents. These changes help clarify the variability across marine systems rather than generalizing all terrestrial systems and all marine systems.

L 48 - 62: However, it can be difficult to distinguish the impacts of gradual (e.g., increasing mean temperatures) and irregular (e.g., increasing storm frequency) climate-induced shifts from changes in underlying naturally stochastic events (e.g., El Niño Southern Oscillation (ENSO)). One such example is the ocean warming phenomenon of marine heatwaves (MHWs). Global oceanic and atmospheric drivers influence regional frequency, duration, and intensity of MHWs^{4,5}, all of which are increasing^{4,6}. In eastern boundary current ecosystems, such as the California Current, MHWs are highly correlated to changes in nutrient availability given the strong correlation between temperature and nutrients^{7,8} (e.g., anomalously high sea surface temperature (SST), low nitrate concentration [NO₃]). MHWs can have notable impacts on coastal ecosystems, such as seagrass beds⁹, coral reefs¹⁰ and kelp forests¹¹, and especially on the foundation species and ecosystem engineers (e.g., seagrasses, corals, kelps) that define these systems. Furthermore, the specific impacts of climate induced changes to these habitat forming sessile organisms

via the coupled impacts of regional non-climate change human influences and species thermal tolerance, greatly increases their vulnerability relative to mobile species¹².

Line 55 Kelp are not just found in upwelling systems. Substantial kelp populations are found in the north Atlantic, southern Australia, temperate east Africa etc – none of these areas are upwelling regions. I therefore suggest the authors rephrase this sentence.

Thank you for catching this. We have removed the reference to upwelling systems only:

L 63 - 65: "Canopy forming kelp species (Order: Laminariales) thrive along temperate rocky coastlines, and are the foundation of productive and species-rich ecosystems that generate a diversity of provisioning, regulating, and supporting ecosystem services³⁰."

Line 57 Suggest replacing geographic with regional
Geographic has been replaced with regional.

Line 58 Delete drivers
This has been deleted.

Line 59 Delete temperate and arctic kelp
This has been deleted.

Line 61 Should be Western Australia as it is the name of an Australia state.
L 67: Thank you, this has been corrected.

Line 72-72 A reference is required to support the statement that sunflower stars are now the key predator of urchins

L 84: We have included a reference to a recent paper from Eisaguirre et al. 2020 (reference number 32). This paper focused on the predator assemblages as predictors of purple urchin densities in the Northern Channel Islands, California. However, they compile sources of current and historical top urchin predators from Alaska to Baja California (Figure 1a). They show that historically in northern California, otters and sunflower stars were the two primary top predators. Because both have been removed from the system (otters were extirpated in the early 1900s and sunflower stars are effectively extinct in the region from SSWS beginning in 2013) there are no top-down controls on urchins in the region.

Line 75-78 An evidence base is required to support the statement that kelp persistence was balanced by top-down urchin predation by sunflower stars

L 89: We have included a reference to Burt et al. 2018 titled 'Sudden collapse of a mesopredator reveals its complementary role in mediating rocky reef regime shifts' (reference number 35). In summary, this study conducted surveys of kelp forest plots that varied in recovery of sea otters and mortality of sunflower stars. By measuring sea otter density, sunflower star density, urchin density and size class, and kelp density across the sunflower star's mass mortality event caused by SSWD, the study showed that the role of otters and sunflower stars maintained kelp forest resilience to alternative state. Though sunflower stars preferred medium and small urchins over large urchins, they played an important complimentary role to sea otters, who preferred larger urchins, in enhancing top down maintenance of a kelp forested state.

Line 84 The authors need to make it clear that such satellite approaches are only

possible for canopy forming kelps such as bull and giant kelp. This technique cannot be used for stipitate or prostrate kelp species.

L 103 - 104: Updated to “Satellite imagery provides a unique perspective on how surface canopy forming kelps respond...”

Line 91-94 I find this a very strong cause and effect statement given the data to hand. I suggest the authors modify the strength of the language to reflect the type of data and analysis that they have undertaken.

We appreciate the reviewer’s suggestion to modify our language to reflect the type of data and analysis we have undertaken. We have restructured and worded the statement:

L 112 – 119: “The persistent lack of kelp from 2014 to 2019 does not appear to be the result of unfavorable environmental conditions alone, but by a combination of unfavorable conditions for kelp productivity (related to warm SST and low nutrients) and conditions favorable for the persistence of urchin populations, including recruitment and low rates of mortality stemming from the absence of predators, disease and starvation. Thus, while fluctuating environmental conditions occurred throughout the past three decades, the combination of abrupt changes in environmental and biological conditions likely hindered the ability of the ecosystem to recover as it had over the past three decades.”

Line 106 Consider adding “even though environmental conditions become more favourable to kelp” after 2019.

Thank you for the suggestion. We have inserted the text, now L 132.

Line 109 It would be useful to have Punta Gorda indicated on the map for those not familiar with the area. While not knowing where Punta Gorda is it would appear that kelp became patchy in this northern region well before 2008, indeed before 1990. Perhaps Punta Gorda is closer to Fort Bragg, but it might be worth the authors discussing this loss of kelp in the far north as this doesn’t seem aligned to the perturbations that are used to explain the decline in later years.

Punta Gorda is just outside (north) of our study region so we have opted to remove the language referring to Punta Gorda and changed the text to:

L 135: “...(north of Fort Bragg)...”

Furthermore, we agree that kelp trends in the northern and southern regions of the study site are likely due to perturbations other than the environmental and biological drivers discussed throughout the manuscript. As was discussed in the original submitted manuscript, we attribute changes to sediment type and poor settlement substrate:

L 136-139: “The northern and southern portions of the historical kelp canopy distribution within our study area are regions characterized by sandy sediment¹⁶ (poor substrate for kelp spore settlement¹⁷), resulting in sparser and patchier distribution than the rockier coastline between Fort Bragg and Jenner prior to the NE Pacific MHW.”

Line 130 Delete and Hs as comes later in the sentence

Thank you, we have corrected this.

Line 139-140 This is a pretty big statement to make when a simple correlative approach has been taken. How can the authors be sure that change in sea urchin numbers is not driven by changing physical conditions.

We thank the reviewer for their insight into sea urchin dynamics in kelp forest ecosystems. We have adjusted the sentence to more accurately represent our findings without extrapolating to predator dynamics in the system.

L 173 - 175: *“Our results revealed that including grazer dynamics in a predictive model more accurately represented sustained low kelp biomass after environmental perturbations from the NE Pacific MHW than the same predictive model with grazer dynamics omitted.”*

Furthermore, we agree with the reviewer that, in part, changes in sea urchin density maybe attributed to changing physical and environmental conditions, such as SST, ENSO, and NPGO (Okamoto et al. 2020, ref. 43 in manuscript). We have included further discussion regarding urchin dynamics:

L 240 - 255: “Regional-scale sea urchin larval recruitment dynamics are associated with large-scale environmental drivers and subsequent population dynamics⁴³. Anomalously high larval recruitment was observed in Fort Bragg, CA, peaking in 2015 and increased larval settlement appeared to be correlated with juvenile and adult urchin densities during the NE Pacific MHW. Although there are no reliable in situ data available for sea urchin densities prior to 2003 from the northern California region, there are purple urchin settlement data as early as 1990 from Fort Bragg (1990-2016⁴³) and Westport, Pt. Cabrillo, and Pt. Arena (1990-1993⁴⁷). These settlement records show there was anomalously high larval settlement from 1993 to 1994 and 1998. Despite bull kelp canopy area being anomalously low between 1995 and 1998 (Fig. 4b), there is no evidence of complete kelp forest collapse or ecosystem shift during that period and no way to verify that high juvenile urchin densities coincided with the high larval densities. Furthermore, anomalously low kelp conditions do not always follow or co-occur with sea urchin larval settlement events (e.g. Fig. 4b 2003 – 2007; Okamoto et al. 2020 Fig. 3a). Given the positive relationship between SST and larval settlement in northern California⁴³ and predictions of more frequent and/or severe MHW⁴⁸, restoration efforts in northern California would benefit from a greater understanding of localized sea urchin population dynamics in that region.”

Line 147-148 I don't know how the reader is supposed to interpret this. The figure merely shows that when urchins are in the model kelp abundance stays low. It doesn't necessarily show that a barren has formed. I suggest restructuring the sentence.

Thank you for your suggestion. We have reworded:

L 183 - 185: *“Including urchin (grazer) dynamics in the PLSR analysis shows that low kelp canopy biomass conditions persist regardless of the anticipated effects of environmental drivers to kelp recovery (Fig. 3d).”*

Line 174-178 Without experimental evidence or reference to the literature this is all conjecture, but is written as if there is evidence of cause and effect. As with other sections I think this needs to be toned down.

Thank you for your suggestion. We have reworded:

L 211 – 216: *“Northern California kelp forests experienced environmental and biological perturbations that likely resulted from the combined effects of (1) the absence of top-down control on urchin populations during and after the NE Pacific MHW (Fig. 4c), (2) abrupt and persistent shifts in SST and nutrient*

conditions across the NE Pacific MHW that were beyond the physiological thresholds of optimum bull kelp growth and reproduction, and (3) an eruption in the population and grazing intensity of the herbivorous purple sea urchin.”

Line 183-187 I don't really understand what message the authors are trying to convey here. I am not clear on what the biological indices were in the MHW. I am also not clear about how giant kelp differed from bull kelp in their responses. I suggest the authors clarify their language.

We have clarified our language in this paragraph by specifically referring to canopy biomass (giant kelp) and canopy area (bull kelp) rather than referring to them together as 'biological indices', as we realize that was confusing. We also worked to (1) clarify what the different responses were by giant and bull kelp as a function of their physiological temperature thresholds and species range limits and (2) describe why bull kelp's response differed due to their sensitivity to abrupt stressors.

L 220 – 233: “However, differences in the expression of kelp forest canopy dynamics between two foundational kelp genera across the NE Pacific MHW highlights that the annual life cycle of bull kelp makes them particularly sensitive to acute stressors³⁶, such as MHWs and prolonged nutrient depletion conditions (Fig. 2 a-c). This is evidenced by the fact that the stepwise decline in northern California bull kelp canopy area across the NE Pacific MHW was not observed in giant kelp (*Macrocystis pyrifera*) canopy biomass at a regional scale in southern California¹⁹ and northern Baja California^{19,20,24}. These observations suggest that giant kelp responded strongly to the NE Pacific MHW as a function of the genera's physiological temperature threshold and latitudinal gradients in SST magnitudes¹⁹, most likely because they were near their southern range and thermal limit in the northern hemisphere (Baja California, Mexico to Aleutian Islands, AK). In contrast, bull kelp forests in our study area, which lie in the middle of their distribution (Point Conception, CA to Unimak Island, AK), did not experience patchy spatial and temporal recovery after the onset of the NE Pacific MHW but maintained very low biomass conditions between 2014 and 2019, perhaps exacerbated by low propagule pressure resulting from patchy, sparse kelp densities and an annual life history strategy³⁶.”

Line 192 I am not clear what the authors mean by endmembers

The endmembers are referring to the northern and southern limits of the study area. However, to help reduce confusion by future readers we have changed the wording and moved the text to a more appropriate section of the results where Figure 1 is described.

L 136-139: “The northern and southern portions of the historical kelp canopy distribution within our study area are regions characterized by sandy sediment⁴⁰ (poor substrate for kelp spore settlement³⁶), resulting in sparser and patchier distribution than the rockier coastline between Fort Bragg and Jenner prior to the NE Pacific MHW.”

Line 202-204 An evidence base to support this statement is required

We have expanded our discussion of why urchin barren conditions are harmful to ecosystem recovery, especially, as in this case, a near complete absence of urchin predators and included four additional references to studies demonstrating this (Smith et al. 2020, Babcock et al. 2010, Eurich et al. 2014, Graham 2004).

L 262 - 272: “In sea urchin barrens, urchins are starved and lack energetic value to predators with high metabolic rates⁵⁰. Moreover, behaviorally mediated predators often track changes in the distribution of profitable prey, which further complicates implications for recovery. Urchin barrens are characterized by low urchin gonad index; because the gonads are what urchin predators target and consume, urchin barren

states potentially limit increased predation of this particular prey⁵¹. The high urchin densities observed in northern California have induced starvation conditions and reduced the nutritional value of the urchins⁵¹. Additionally, since the overall ecosystem biodiversity of urchin barrens is severely reduced⁵², opportunities for predator (otters, sunflower stars, etc.) recovery is diminished. Ecosystem recovery is further limited by evidence that the effects on prey can lag behind the recovery of a predator⁵³.”

Smith, J. G. *et al.* Behavioral responses across a mosaic of ecosystem states restructure a sea otter-urchin trophic cascade. *Proc. Natl. Acad. Sci.* (2020).

Eurich, J. G., Selden, R. L. & Warner, R. R. California spiny lobster preference for urchins from kelp forests: implications for urchin barren persistence. *Mar. Ecol. Prog. Ser.* **498**, 217–225 (2014).

Graham, M. H. Effects of Local Deforestation on the Diversity and Structure of Southern California Giant Kelp Forest Food Webs. *Ecosystems* **7**, 341–357 (2004).

Babcock, R. C. *et al.* Decadal trends in marine reserves reveal differential rates of change in direct and indirect effects. *Proc. Natl. Acad. Sci.* **107**, 18256–18261 (2010).

Line 211-212 I am not sure how the previous sentence leads the reader to this conclusion. I suggest the authors make this connection more clearly.

We have attempted to clarify our wording.

L 286 - 290: “Evidence suggests that the pathogen associated with SSWS in the sunflower star was not temperature dependent, nor responsible for disease observed in other asteroids throughout the region⁵⁴. This may explain why recovery for sunflower stars across the region, and in turn kelp forest recovery in northern California, remain absent despite temperature and nutrient conditions recovering slightly in 2017.”

Fig 2d Given the amount of temporal site to site variation in kelp density I am not sure it is appropriate to amalgamate across the whole study region. Or perhaps the authors just need to justify this approach in the methods. In the figure legend the authors need to state that orange is where the blob and the el nino even overlap.

The authors agree that it can be difficult to detect long term trends at regional scales (>100km). Bell et al. 2020 argued that most declines happen at a local scale and are related to local scale processes. However, the declines presented in this study are unique in that a regional decline occurred across over >350 km of coastline, as Figure 1b demonstrates. Therefore, we believe that it is best to look at these data at a regional scale in the context of kelp dynamics and the environmental and biological drivers that occurred across the NE Pacific MHW. We have included a sentence to the Methods to help the reader understand the reason for the scale at which this study was conducted.

L 334 - 338: “Although most kelp decline occurs at small scales driven by local processes (Bell et al. 2020), losses of northern California bull kelp in 2014 occurred across nearly 350 km of continuous coastline (Fig. 1b). Therefore, the environmental and biological drivers of kelp were investigated in the context of regional scale kelp dynamics that occurred across the NE Pacific MHW (S5 and S6).”

Furthermore, we have corrected the figure 2 caption text (L 150 – 151) to reflect orange as the overlapping ‘blob’ and el nino year.

Fig 3 From the methods it would appear that the authors have not checked for

collinearity between physical predictors. It is likely that many of these will be correlated and therefore it may not be appropriate to have them all in the model. The acronyms need to be defined in the figure legend.

Yes, the authors agree that some of the physical variables are multi-collinear, notably seasonal sea surface temperature (SST), seasonal nitrate (NO_3) and pacific decadal oscillation (PDO). This collinearity was an important consideration when selecting the partial least squares regression (PLSR) approach. To help clarify both our statistical approach and the inherent characteristics of the dataset, we have added language to the methods section that expand on the benefits of PLSR for this type of dataset (L 380 - 399) and supporting figures in the supplemental material (S8 and S9). Supporting supplemental material includes (1) a correlation matrix of all environmental and biological variables used in the PLSR analysis and their respective pairwise Pearson correlation coefficient and pairwise kernel density distribution (S8), and (2) mean squared errors matrix from a cross-validation analysis of environmental variables (S9a) and combined environmental and biological variables (all indices; S9b). Furthermore, we describe how we selected the variables presented in each of the PLSR configurations in detail in the methods.

L 380 - 399: “Statistics and Reproducibility: **determining the drivers of bull kelp canopy coverage.** Following determination of maximum annual kelp canopy coverage, a partial least squares regression¹³ (PLSR) was used to investigate the temporal response of kelp canopy coverage in northern California (Mendocino and Sonoma Counties; Fig. 1a) to large- and local-scale oceanographic and biological processes. PLSR combines principle component analysis (PCA) and multiple linear regression to maximize covariance between the predictor and response variables. This method works particularly well where (1) strong collinearity between predictor variables occurs (**S8**) and (2) a low number of observations would normally reduce model performance⁶⁶. Many of the variables used in this study present strong multi-collinearity (**S8**) especially between seasonal SST, seasonal NO_3 and PDO.

PLSR analysis was conducted using the PLSRegression function in the Python 3.7 sklearn.cross_decomposition machine learning statistical module. Using a k-fold cross-validation technique, environmental variables were selected by calculating the mean squared error (MSE) and determining the optimal configuration via the lowest MSE (**S9a**). The cross-validation showed that the number of predictor variables had little influence on the performance of the environmental-only indices' first component (**S9a**). Therefore, a one component, 9 variable configuration was selected. Although a two component, 4 variable configuration was optimal for environmental and biological indices combined (**S9b**), the study's goal was to compare how the 'environmental-only' model results changed when adding biological forcing (purple urchins). As a result, a one component, 10 variable configuration was selected.”

To address the reviewer's comment regarding acronyms, we have added acronym definitions to Figure 3 legend and a detailed description of how each environmental variable can influence kelp canopy dynamics in the Supplemental material.

Figure 3, L 168 - 172: “Predictor variable acronyms are as follows: purple urchin density - 'Purple Urchin'; seasonal nitrate concentrations - 'Summer NO_3 ' and 'Spring NO_3 '; marine heatwave days - 'MHW Days'; seasonal sea surface temperature – 'Summer SST' and 'Spring SST'; mean significant wave height – 'Mean H_s '; Pacific Decadal Oscillation – 'PDO'; North Pacific Gyre Oscillation – 'NPGO'; Multivariate El Niño/Southern Oscillation Index – 'MEI'. See Methods for detailed description of how each environmental variable influences kelp canopy dynamics.”

Supplemental Material (S5) (reference numbered in order of appearance in supplemental material):

“SST Index –SST conditions effect the distribution (physiological temperature threshold), gametophyte maturation⁴, and the seasonal growth rates⁵.

NO_3 Index – Nitrate conditions fuel growth seasonally. Growth rates are primarily high in the spring and early summer due to the availability of nutrient rich water brought to the surface by seasonal upwelling. Growth rates are generally low in the summer due to limited nitrate conditions^{5,6}.

H_s Index – Bull kelp are an annual algal species and in exposed regions, such as the northern California coast, are typically removed by strong wave forces during fall and winter storms. Therefore, seasonal and annual trends in significant wave height influence canopy distribution⁶.

MEI Index – the Multivariate El Niño/Southern Oscillation (ENSO) Index (MEI.v2) is indicative of global climate disruptions and derived from five different variables (sea level pressure, sea surface temperature, zonal and meridional components of the surface wind, and outgoing longwave radiation). Disruptions to oceanographic conditions via ENSO patterns influence SST, NO₃, and wave height conditions (H_s). Studies have found ENSO to be an important driver of kelp dynamics across the globe⁷⁻¹¹.

NPGO Index – the North Pacific Gyre Oscillation is an oceanic climate index derived from the second mode of sea surface height variability in the northeast Pacific and influences sea surface nutrient dynamics in the North Pacific Gyre and California Current. Many studies in the NE Pacific have found NPGO to be an important driver of regional kelp dynamics^{10,12,13}.

PDO Index – the Pacific Decadal Oscillation index is derived from the first mode of sea surface temperature variability in the north Pacific poleward of 20°N. Many studies in the NE Pacific have found PDO to be an important driver of regional kelp dynamics^{10,12,13}.

Fig 4 Add P to LSR in the figure legend

Figure 4 does not show any PLSR results, rather only the least squares regression fits (both linear and polynomial) for panels b - e. Therefore, we have not made the suggested change, but have attempted to clarify the application of the ordinary and polynomial least squares regression fits (L 206 - 207).

S3 The authors should state in the legend that the grey cells represent significant relationships.

Now S4: Thank you. We have clarified that the shading represents a significant relationship in the table legend.

Reviewer #3 (Remarks to the Author):

Review of 'Oceanographic and ecological perturbations associated with a marine heatwave dramatically shift the structure of a temperate coastal marine ecosystem' by McPherson et al

This paper describes the collapse and conversion of large bull kelp forests, and decreases in sea star populations in Northern California to urchin dominated barrens. The paper is well written, and the findings are clearly important to show.

My main concern is (probably not surprising) that two of the Authors recently told the same general story in Scientific Report (2019, 9:15050) and that the novelty therefore is minimal. Indeed, I see no new ecological take home messages when comparing the abstracts between the 2019 scientific report paper and the present manuscript. I am, of course, aware that there are methodological differences between the papers – and that the time series data are longer for this new paper – but that does very little to change the overall impression about the findings.

So the fate of this manuscript very much lies with the editor – i.e., how ‘novel’ should the results be before a paper can be published in this particular journal.

We would like to thank the reviewer for providing useful comments and improving the quality of this manuscript. However, the authors strongly disagree with the reviewer’s comment on the novelty of this paper and on the similarities with the Scientific Reports publication. For such a widespread environmental (MHW) and ecological (kelp forest to urchin barren along 350 km of coastline) event, the lack of literature available is concerning to scientists and managers of kelp forest ecosystems worldwide. Rather than the qualitative approach taken in the Scientific Reports publication, we believe that this study sets a quantitative precedent for understanding the drivers of environmental and biological processes across the entire region (both long term and across the NE Pacific MHW). This approach is especially important as kelp forests become more vulnerable to changing environmental conditions associated with climate change.

We provide evidence of this study’s novelty through four key differences to the Scientific Reports publication. First, the lead author of this manuscript, who was responsible for conceptualization and methodology, and who was solely responsible for writing the original draft of this manuscript – was not an author on the Scientific Reports publication. Second, we have presented the largest bull kelp record available (34 years temporally and spatially) from northern California, a region that is historically under-sampled with regard to aerial surveys and in situ data collection. As the foundation species of the entire northern California ecosystem, the addition of this time series to the scientific community is invaluable and was notably lacking from the Scientific Reports publication, which only presented limited spatial and temporal coverage of bull kelp. Third, we have done so while applying novel methods for remote sensing of bull kelp and have combined this perspective with a quantitative analysis of biological and environmental variables across the 34 year time series (including important temperature-nutrient dynamics). Again, none of these drivers were quantitatively assessed in the Scientific Reports article referenced, rather qualitatively discussed. Fourth, the authors have included a forecasting approach to better glean what environmental and biological changes would lead to (or not lead to) a recovery in bull kelp. This approach leads to an essential understanding of the role that drivers may play in recovery through time and allows us to assess the impacts of both top-down (biological) versus bottom-up (physical and environmental) process on bull kelp dynamics and recovery.

Below I outline a few minor comments in chronological order.

L32-34. It sounds like the satellite images can be used to detect declines in sea stars (which they clearly cannot – only kelp cover)

Thank you for the suggestion. We have edited for clarity:

L 32 - 34: “Using a suite of in situ and satellite derived data, we demonstrate that the abrupt ecosystem regime-shift initiated by a multi-year MHW was preceded by declines in keystone predator population densities.”

L72-73. Are there no other natural urchin predators than otter? No fish that will eat the urchin?

Eisaguirre et al. 2020 (reference number 32) compiled sources of current and historical top urchin predators from Alaska to Baja California (Figure 1a). They show that historically in northern California, otters and sunflower stars were the two primary top predators. Because both have been removed from the system (otters were extirpated in the early 1900s and sunflower stars are effectively extinct in the region from SSWS beginning in 2013) there are no top-down controls on urchins in the region. Other predators such as sheephead fish and spiny lobster are important mesopredators of urchins in southern

California, but there is no evidence other species playing as prominent a role as sunflower stars in northern California prior to SSWD. Wolf eels and red rock crab will prey on urchins, but they are very rare in the system and therefore play a small role in controlling urchin dynamics. We now cite this paper in Line 90.

Eisaguirre, J. M., Davis, K., Carlson, P. M., Gaines, S. D. & Caselle, J. E. Trophic redundancy and predator size class structure drive differences in kelp forest ecosystem dynamics. *Ecology* **101**, 1–11 (2020).

L85. You say that satellite images are great to monitor kelp – however, its only really useful for kelp with surface canopies, i.e. *Macrocystis* and *Nereocystis* – most kelp species cannot be detected with satellites.

Thank you for the suggestion. We have edited for clarity:

L 103 - 105: "Satellite imagery provides a unique perspective on how surface canopy forming kelp respond to both acute climate manifestations and in situ biological trends leading to ecosystem phase shifts and can compensate for the scarcity of historical kelp data in northern California."

L90 – this again sounds like the satellite images can be used to detect declines in sea stars (which they clearly cannot – only kelp cover)

Thank you for the suggestion. We have edited for clarity:

L 108 - 112: "Using a 34-year timeseries of kelp canopy coverage derived from United States Geological Survey (USGS) Landsat imagery in combination with large-scale, local-scale, and biological drivers, we infer that historically declining predator densities might have laid the groundwork for the observed ecosystem phase shift in northern California initiated by a multi-year MHW."

L94-96. How does your work provide 'context for monitoring'? - as I said, most kelp systems cannot be monitored with satellites.

We have clarified the statement so that 'context for monitoring' aligns more specifically with what our study focuses on, surface canopy forming kelp and their associated environmental and biological drivers. We have modified the statement to reflect this:

L 119 - 122: This work provides context for monitoring of long-term biological trends and environmental response in surface canopy forming kelp forest ecosystems globally where satellite monitoring can be applied. These techniques are becoming increasingly important for designing adaptive management strategies to mitigate the impacts of long-term and abrupt environmental stressors.

Somewhere in the introduction: I am missing a bit of natural history of *Nereocystis*. In particular that it only lives to 12-18 month (which is pretty unusual for a very large kelp) – so that there are very large phenological short term and small scale fluctuations in canopy cover.

*Thank you for your suggestion to add more detailed information about *Nereocystis*. We agree that it adds value to the manuscript.*

L 90 - 96: "In contrast to giant kelp, which can live for many years and continuously produce new reproductive and vegetative fronds, bull kelp's annual life history is limited to the production of a single stipe and its reproductive blades in its lifetime^{36,37}. As such, mechanisms for spore dispersal are limited to a narrow window between the maturity of the kelp and the onset of fall and winter storms, which usually

dislodge adult bull kelp from the substrate (except in areas protected from wave energy). These factors lead to high spatial and temporal variability in the distribution and abundance of the surface canopy that can be observed through remote sensing techniques.”

L145-147. This could be a place where you discuss problems associated with low propagule pressure (as another reason why recovery is slow). You spend a lot of time discussing urchins and top-down control as a future limitations. I think you should include more discussion on how few propagules are now being released to the system – and that, by itself, limit recovery.

We have addressed this concern in both the introduction (previous comment, L 90 - 96) and briefly in the discussion. Rather than discussing specific factors between low propagule pressure and slow recovery, we provide context to the life history strategies and limitations on spore dispersal and when comparing response and recovery patterns between bull kelp and giant kelp.

L219 – 235: *“Co-varying environmental parameters, including SST and nitrate concentrations, historically maintained fluctuating yet stable long-term trends of bull kelp conditions in northern California (Fig. 4d; $p > 0.05$). However, differences in the expression of kelp forest canopy dynamics between two foundational kelp genera across the NE Pacific MHW highlights that the annual life cycle of bull kelp makes them particularly sensitive to acute stressors³⁶, such as MHWs and prolonged nutrient deplete conditions (Fig. 2 a-c). This is evidenced by the fact that the stepwise decline in northern California bull kelp canopy area across the NE Pacific MHW was not observed in giant kelp (*Macrocystis pyrifera*) canopy biomass at a regional scale in southern California¹⁹ and northern Baja California^{19,20,24}. These observations suggest that giant kelp responded strongly to the NE Pacific MHW as a function of the genera’s physiological temperature threshold and latitudinal gradients in SST magnitudes¹⁹, most likely because they were near their southern range and thermal limit in the northern hemisphere (Baja California, Mexico to Aleutian Islands, AK). In contrast, bull kelp forests in our study area, which lie in the middle of their distribution (Point Conception, CA to Unimak Island, AK), did not experience patchy spatial and temporal recovery after the onset of the NE Pacific MHW but maintained very low biomass conditions between 2014 and 2019, perhaps exacerbated by low propagule pressure resulting from patchy, sparse kelp densities and an annual life history strategy³⁶.”*

L155. The only p-value in the entire paper. When I compare to the method section it is not clear what test you used, what assumptions was required or why you did this particular test – and why you did not do more tests)

Please see our response below addressing this reviewer’s comment.

L166. Here is an example on means and slope values – but without p-values – so why is this a different approach compared to l155?

The authors thank the reviewer for pointing out inconsistencies in the presentation of important statistical information in the two comments above. While the p-values might not have been previously reported clearly in the text or Figure 4, all the relevant statistical information for temporal trends and significance values were contained in the supplemental material (now S4) in the original submitted draft and readers were directed to it in the text and Figure 4 legend. As the reviewer requested, we have corrected inconsistencies in the p-value presentation in the text and added a brief paragraph to the Methods explaining the correlative approach displayed in the text (L 221), Figure 4 caption, S3, and S4.

L 400 - 409: *“After determining and modeling important drivers of kelp canopy area, a least squares regression (LSR) approach was utilized to understand significant and insignificant temporal changes in*

relevant biological and environmental indices (kelp canopy area, spring NO₃, MHW Days, sunflower star density, and purple urchin density) across the entire timeseries, prior to (pre-2014) the NE Pacific MHW, and following (post-2013) the NE Pacific MHW. For all indices, except the pre-2014 sunflower star index (which was optimized to a polynomial LSR), ordinary LSRs were fit to data. This simple correlative approach was valuable for understanding how these relevant variables changed on both long and short-term timescales with relevance to dramatic declines in kelp canopy coverage. Trendline error (S3) and regression statistics (S4; slope, r^2 , and p-value) are presented in the supplemental material.

L183-184. Its interesting that *Macrocystis* was less affected by urchins/mhw/ssws. It would be nice to know how Macro and *Nereo* have different temperature sensitivities and what there latitudinal ranges are.

We have edited the paragraph to include a more nuanced discussion of the two species temperature distributions and ranges for context. We have also added additions to Figure 2 showing the physiological temperature threshold for bull kelp (17°C; Vadas 1972) and NO₃ deplete conditions (13.1°C; Garcia-Reyes et al. 2014).

L 227 - 235: “These observations suggest that giant kelp responded strongly to the NE Pacific MHW as a function of the genera’s physiological temperature threshold and latitudinal gradients in SST magnitudes¹⁹, most likely because they were near their southern range and thermal limit in the northern hemisphere (Baja California, Mexico to Aleutian Islands, AK). In contrast, bull kelp forests in our study area, which lies in the middle of their distribution (Point Conception, CA to Unimak Island, AK), did not experience patchy spatial and temporal recovery after the onset of the NE Pacific MHW but maintained very low biomass conditions between 2014 and 2019.”

L202-205. I would question that lower nutritional values in urchins limits recovery of predators. From your photos it looks in scientific report it looks like there are so many urchins that are easy to pick for predators that a slightly smaller nutritional value is offset by a much lowered searching/catching/handling time.

We have improved clarity on this topic by expanding our discussion of why urchin barren conditions are harmful to ecosystem recovery and noting that predator behavior and foraging strategy is species specific and sometimes specific to the individual (eg. otters). However, we still argue for the overall low nutritional value of urchins in northern California. Furthermore, we argue that predator recovery will be challenging despite large urchin populations via three new references (Smith et al. 2020, Eurich et al. 2014, Babcock et al. 2010, and Graham 2004).

L 259 - 272: “Throughout California, a suite of predators (e.g. sunflower stars, sea otters, California Sheephead fish³⁴), and their complementary effects, play an essential role in maintaining stable forested states by enhancing resiliency via size-dependent predation⁴⁴, even when environmental perturbations occur. In sea urchin barrens, urchins are starved and lack energetic value to predators with high metabolic rates⁵⁰. Moreover, behaviorally mediated predators often track changes in the distribution of profitable prey, which further complicates implications for recovery. Urchin barrens are characterized by low urchin gonad index; because the gonads are what urchin predators target and consume, urchin barren states potentially limit increased predation of this particular prey⁵¹. The high urchin densities observed in northern California have induced starvation conditions and reduced the nutritional value of the urchins⁵¹. Additionally, since the overall ecosystem biodiversity of urchin barrens is severely reduced⁵², opportunities for predator (otters, sunflower stars, etc.) recovery is diminished. Ecosystem recovery is further limited by evidence that the effects on prey can lag behind the recovery of a predator⁵³.”

- Smith, J. G. *et al.* Behavioral responses across a mosaic of ecosystem states restructure a sea otter-urchin trophic cascade. *Proc. Natl. Acad. Sci.* (2020).
- Eurich, J. G., Selden, R. L. & Warner, R. R. California spiny lobster preference for urchins from kelp forests: implications for urchin barren persistence. *Mar. Ecol. Prog. Ser.* **498**, 217–225 (2014).
- Graham, M. H. Effects of Local Deforestation on the Diversity and Structure of Southern California Giant Kelp Forest Food Webs. *Ecosystems* **7**, 341–357 (2004).
- Babcock, R. C. *et al.* Decadal trends in marine reserves reveal differential rates of change in direct and indirect effects. *Proc. Natl. Acad. Sci.* **107**, 18256–18261 (2010).

L224-227. Again, are there no other predators than otters? Here it is related to management – is otter really the only ‘natural’ option for urchin control?

Again, both otters and sunflower stars are important historical predators of urchins in northern California. Wolf eels and red rock crab will prey on urchins, but they are very rare in the system and therefore play a small role in controlling urchin dynamics.

L261-263. Here you make a simple reference to another paper that describe the sampling methods related to sea stars and urchins. However, these biological data are really important so the reader should know much more about the methods; how many samples, where, when and at what depth? Without such standard information readers cannot evaluate the strength the trends.

We thank the reviewer for pointing out the lack of detail in the presentation of methods regarding subtidal biological data collection and have worked to add details to both the Methods and the supplemental material that will help readers know where, when, and at what depth these samples were collected. We utilized two sources for our subtidal biological data (Reef Check California and California Department of Fish and Wildlife). We have described the detailed methodology used by each organization in the Methods section as follows:

*L 355 - 379: Biological indices, including purple sea urchin (*Strongylocentrotus purpuratus*) and sunflower star (*Pycnopodia helianthoides*) densities, were obtained from California Department of Fish and Wildlife (CDFW; 2003 - 2018) and Reef Check California (2007 – 2018)²⁸ subtidal rocky reef habitat surveys (**S6**). Both organizations conduct annual surveys through the summer and early fall in northern California. Reef Check California utilizes trained citizen scientists to support coastal ecosystem monitoring, management, and to promote stewardship of sustainable kelp forest communities. Reef Check California surveyed 27 sites in the northern California Sonoma and Mendocino counties between 2007 and 2018 (**S7**) ranging between 7 and 22 sites annually. Each site consisted of 2 depth strata (inshore: 0 – 10 m and offshore: 10 – 20 m) and 6, 60 m² (2 x 30 m) invertebrate and algal transects. Three 60 m² transects were conducted in each depth strata generally parallel to shore. Partial transects densities were calculated when more than 50 individuals of a species were counted along a distance of at least 5 m.*

*CDFW subtidal surveys occur as part of the agency’s kelp ecosystem management program²⁹. CDFW surveyed 12 sites in the northern California Sonoma and Mendocino counties between 2003 and 2018 (**S7**) ranging between 2 and 11 sites annually. Random transects were placed within four depth strata (0 – 4.5 m, 4.5 m – 8.3 m, 9 – 13.7 m, and 13.7 – 18.3 m), divided equally within the full depth range (0 – 18 m) with 60 m² transects (2 x 30 m). Transect locations were at predetermined random GPS coordinates greater than 70 m apart and generally parallel to shore. At each site 15–55 transects were surveyed, with equal numbers of transects per depth stratum. All organisms were counted and recorded within the transect area regardless of density distribution.*

Annual densities for the entire study region were determined by taking the mean of all 60 m² transects conducted by Reef Check California and CDFW. Standardized indices were calculated by

removing the long-term annual climatology from absolute annual means and normalizing to the standard deviation.”

The supporting figure presented in the supplemental material (S7) visualizes the temporal and spatial distribution of site coverage and transect area. Transect area is visualized temporally (S7a) and spatially (S7b and c) by year and site, respectively, for both data sources. Panel S7c shows the distribution of sites within the study area. Although there are portions of the coast where survey sites are clustered and areas where survey sites are notably absent, site selection is based on several factors including accessibility, kelp presence/absence, substrate type, and locations of historical abalone recreational fishing grounds. Between the two organizations, nearly 160,000 m² of benthos was surveyed between 2003 and 2018.

Somewhere in the method section. I would like to see some clear descriptions of what statistical tests you used, why and how test assumptions are addressed.

We thank the reviewer for the suggestion to include more details on our statistical model selection and have included more language into the Methods:

L 384 - 389: “PLSR combines principle component analysis (PCA) and multiple linear regression to maximize covariance between the predictor and response variables. This method works particularly well where (1) strong collinearity occurs between predictor variables occurs (**S8**) and (2) a relatively low number of observations would otherwise reduce model performance⁶⁶. Many of the variables used in this study present strong multi-collinearity (**S8**) especially between seasonal SST, seasonal NO₃ and PDO.”

Furthermore, the selection of PLSR is warranted and validated by Carrascal et al. (2009) (reference number 66), who carried out a simulation experiment to compare PLSR with multiple regression (a type of GLM) and with a combination of PCA and multiple regression. They varied the number of predictor variables and sample sizes. Although the PLSR models explained a similar amount of variance to those results obtained by the other techniques, they were more reliable than other techniques when identifying relevant variables and their magnitudes of influence, especially in cases of small sample size and low tolerance.

I would also like to see some sort of variability associated with some of the graphs.

Currently no data variability is shown so I have no idea about how robust your data are. I would also like to know about sample sizes in the figure legend (scientific figures + legends should preferably show some kind of central values, some kind of data dispersion and the associated replication levels).

With respect to the two comments above regarding data variability, we have made changes to Figure 2, added additional supplemental material for Figure 4 trendlines (S3), and created a summary box plot that contains data distribution for all environmental and biological variables (S6).

In Figure 2 (a – c), we have included a shaded grey area around the standardized long-term mean SST distribution representing \pm SD. Based on the addition to the plot, it is apparent that NE Pacific MHW SST distributions were outside even the long-term mean \pm SD distribution.

In Figure 2d, error estimates on kelp canopy area were calculated using the normalized root mean squared error (NRMSE) based on work by Finger et al. in review. This error estimate is based on the relationship between California Department of Fish and Wildlife (CDFW; 2 m resolution) and USGS Landsat kelp canopy area (30 m resolution) across 9 years of overlapping data from each sensor. The RMSE of the relationship was normalized to the range in CDFW kelp canopy area and therefore estimates the fraction of Landsat derived kelp canopy error relative to CDFW aerial flyover survey kelp

canopy area for each year (NRMSE = 0.176). Brief explanation of error distribution and kelp error methodological details have been added to the Figure 2 legend text (“Annual error estimates for kelp canopy area are represented by the black error bars based on work by Finger et al. in review.”).

To help readers gain a more thorough picture of the variability and significance of regression estimates for environmental and biological variables preceding and after the NE Pacific MHW, combined trendline and error estimates for Figure 4 LSRs were added to the supplemental material (S3). Readers are directed to the supplemental material in the Figure 4 legend text.

Finally, since all the variables are presented in standardized indices, many of the plots already inherently display the error distribution of the dataset. The standardized indices are calculated by normalizing the absolute anomalies to the entire dataset’s standard deviation. However, we agree that it is important for the readers to understand what the absolute data distribution is for each variable. Therefore, the box and whisker plot is a valuable addition to the supplemental material (S6) and includes the median, 0.25 and 0.75 quartile limits, minimum value, maximum value, and outliers for each variable. Variables are also classified into their sampling frequency (annual, monthly, daily, and hourly).

I also think the ecological context and discussion could benefit from invoking trophic cascades and alternative stable state concepts/theories (which are the dominant ecological processes going on here – but none of these classic ecological concepts are mentioned in the paper).

We thank the reviewer for the suggestion to include a discussion of the ecological concepts relevant to this study. However, given the controversial nature of alternative stable state theory we refrain from strictly defining the ecological processes occurring in the northern California region at this time. However, we provide context and a more nuanced discussion that, we believe, provides sufficient environment to alternative stable state theory in relation to this study, while also recognizing that we have limited observations to define the current urchin barren state as such. We have included text in both the introduction and discussion sections.

L 74 – 76: “These shifts between alternative stable states of these kelp forests often reflect cascading interactions across trophic levels through bottom-up (i.e. environmental influences on kelps) and top-down²⁷⁻²⁹ (i.e. changes in predator control of grazers) processes.”

L 273 – 282: “Despite potential limitations of urchin barrens on predator recovery, reintroduction of sea otter populations into urchin barrens has resulted in phase shifts back to forested states in some locations (e.g. Aleutian Islands²⁶). It is unclear from this analysis what future phase state dynamics will occur with the reintroduction of a top predator given the strong potential that this urchin barren constitutes a kelp forest alternative stable state²⁶. Although we refer to the recent wide-spread kelp forest loss as a phase shift and cannot currently provide proof of a true kelp forest alternative stable state, Filbee-Dexter and Scheibling 2014 argues that in most cases the formation of urchin barrens can be regarded as such. Considering that the dynamics of the wide-spread urchin barren in northern California has similar patterns to other urchin barrens, hysteresis (discontinuous phase shift) and strong positive feedbacks may maintain the current state for a prolonged period of time.”

REVIEWERS' COMMENTS:

Reviewer #3 (Remarks to the Author):

The authors have done a great job addressing my (and the other two reviewers) comments. The manuscript is much improved and will be an important contribution to research about kelp forest ecology and impacts from extreme events.

I have not further comments – except a few minor details related to the new text.

L 37-38. Low PRIMARY productivity (note that secondary productivity may be higher in the new state because of all the urchins – however PP should be lower because seaweed are virtually gone)

L153. Its not entirely clear what the error bars are – standard errors I assume?

L202. It is not clear what the values in yellow refers to and what 'm' means

L268 change 'the nutritional value of the urchins' to 'nutritional value' (we already know its about urchins from the first part of the sentence

L387 delete 'occurs' (its already used in the beginning of the sentence)

We thank the reviewer for thoroughly reviewing and reading the manuscript and taking the time to add final touches. Please see specific line changes below.

Reviewer #3 (Remarks to the Author):

The authors have done a great job addressing my (and the other two reviewers) comments.

The manuscript is much improved and will be an important contribution to research about kelp forest ecology and impacts from extreme events.

I have no further comments – except a few minor details related to the new text.

L 37-38. Low PRIMARY productivity (note that secondary productivity may be higher in the new state because of all the urchins – however PP should be lower because seaweed are virtually gone)

L38: Thank you for pointing out this important distinction. We have included the word ‘primary’ into the sentence.

L153. Its not entirely clear what the error bars are – standard errors I assume?

L569 - 571: As we mentioned in our previous reviewer response document, Figure 2d error estimates on kelp canopy area were calculated using the normalized root mean squared error (NRMSE) based on work by Finger et al. 2021. This error estimate is based on the relationship between California Department of Fish and Wildlife (CDFW; 2 m resolution) and USGS Landsat kelp canopy area (30 m resolution) across 9 years of overlapping data from each sensor. The RMSE of the relationship was normalized to the range in CDFW kelp canopy area and therefore estimates the fraction of Landsat derived kelp canopy error relative to CDFW aerial flyover survey kelp canopy area for each year (NRMSE = 0.176). We expanded on our brief explanation of error distribution and kelp error methodological details from the previous submission to help readers understand the derivation of the error bars and reference Finger et al. 2021.

“Annual error estimates (black error bars) for kelp canopy area were determined using the normalized root mean square error (NRMSE) between CDFW aerial flyover surveys and USGS Landsat imagery⁶⁷.”

L202. It is not clear what the values in yellow refers to and what ‘m’ means

L173 - 176: The ‘m’ denoted slope, but we realized that was confusing next to the units of meters (m) and have specifically indicated that it is the slope. We have also moved where we reference the slope value within the sentence to indicate relevance of the trend in purple sea urchin density to the ‘widespread shift in purple urchin foraging behavior.’

L268 change ‘the nutritional value of the urchins’ to ‘nutritional value’ (we already know its about urchins from the first part of the sentence)

L234 - 235: Thank you, we have made this edit to the sentence.

L387 delete 'occurs' (its already used in the beginning of the sentence)

L353: Thank you, we have made this edit to the sentence.